# An engineered CRISPR-Cas12a variant and DNA-RNA hybrid guides enable robust and rapid COVID-19 testing

Kean Hean Ooi[1,2,8], Mengying Mandy Liu [1,2,8], Jie Wen Douglas Tay[1,2,3,8], Seok Yee Teo[1,2,3,8], Pornchai Kaewsapsak[2,8], Shengyang Jin[3], Chun Kiat Lee [4], Jingwen Hou[5], Sebastian Maurer-Stroh [6], Weisi Lin[5], Benedict Yan[4], Gabriel Yan [7], Yong-Gui Gao[3] & Meng How Tan [1,2✉]

Extensive testing is essential to break the transmission of SARS-CoV-2, which causes the ongoing COVID-19 pandemic. Here, we present a CRISPR-based diagnostic assay that is robust to viral genome mutations and temperature, produces results fast, can be applied directly on nasopharyngeal (NP) specimens without RNA purification, and incorporates a human internal control within the same reaction. Specifically, we show that the use of an engineered AsCas12a enzyme enables detection of wildtype and mutated SARS-CoV-2 and allows us to perform the detection step with loop-mediated isothermal amplification (LAMP) at 60-65 °C. We also find that the use of hybrid DNA-RNA guides increases the rate of reaction, enabling our test to be completed within 30 minutes. Utilizing clinical samples from 72 patients with COVID-19 infection and 57 healthy individuals, we demonstrate that our test exhibits a specificity and positive predictive value of 100% with a sensitivity of 50 and 1000 copies per reaction (or 2 and 40 copies per microliter) for purified RNA samples and unpurified NP specimens respectively.

[1] School of Chemical and Biomedical Engineering, Nanyang Technological University, Singapore, Singapore. [2] Genome Institute of Singapore, Agency for Science Technology and Research, Singapore, Singapore. [3] School of Biological Sciences, Nanyang Technological University, Singapore, Singapore. [4] Department of Laboratory Medicine, National University Hospital, National University Health System, Singapore, Singapore. [5] School of Computer Science and Engineering, Nanyang Technological University, Singapore, Singapore. [6] Bioinformatics Institute, Agency for Science and Technology and Research, Singapore, Singapore. [7] Division of Infectious Diseases, Department of Medicine, National University Hospital, National University Health System, Singapore, Singapore. [8] These authors contributed equally: Kean Hean Ooi, Mengying Mandy Liu, Jie Wen Douglas Tay, Seok Yee Teo, Pornchai Kaewsapsak. ✉email: mh.tan@ntu.edu.sg

COVID-19 is an ongoing global pandemic caused by SARS-CoV-2, a novel coronavirus of zoonotic origin. The outbreak was first reported in Wuhan, China[1–3] and has since spread to more than 200 countries. As of 3 December 2020, there are over 64.5 million confirmed cases and 1.5 million deaths worldwide, underscoring the severity of the disease.

Given the high human-to-human transmission potential of SARS-CoV-2 including from asymptomatic carriers[4–6], rapid and accurate diagnosis is critical for timely treatment and outbreak control. Currently, quantitative real-time PCR (qRT-PCR) is the gold standard method to detect COVID-19. However, it requires specialized and expensive instrumentation to run and thus must be carried out in dedicated facilities with the necessary equipment and expertise. Furthermore, the turnaround time for qRT-PCR is too slow. Even excluding the time it takes to transfer samples from collection points to the test facilities, the PCR process itself typically requires at least 1.5 h to run. Hence, there is a demand for rapid point-of-need tests that can identify infected individuals more quickly. Different types of rapid diagnostic tests are actively under development or have already been rolled out, including serological tests that detect human antibodies against SARS-CoV-2 and antigen tests that detect presence of viral proteins. However, the former have limited practical use for identifying infectious individuals as antibodies are only detectable in later stages of infection when opportunities to treat and limit disease transmission have passed, while the latter suffer from poor sensitivity. Another type of rapid diagnostic tests involves isothermal amplification methods, such as recombinase polymerase amplification (RPA)[7] and loop-mediated isothermal amplification (LAMP)[8]. However, such methods are challenging to implement well because they often generate spurious non-specific products that yield false positive results.

CRISPR-Cas has emerged as a powerful technology that can potentially drive next-generation diagnostic platforms. After binding to a specific target substrate, certain Cas enzymes are then hyperactivated to cleave all neighbouring nucleic acids indiscriminately[9–11]. By programming the Cas nuclease to recognize desired sequences, such as those containing cancer mutations or from pathogens-of-interest, and providing single-stranded DNA (ssDNA) or RNA reporter molecules in the reaction mix, various groups have successfully developed CRISPR-based diagnostics (CRISPR-Dx) for a range of applications[11–16]. Unsurprisingly, it has also not escaped attention that the same technology can be applied to tackle the COVID-19 outbreak. Within a few months, multiple CRISPR-based assays for the disease have been announced (Supplementary Data 1)[17–40], underscoring the ease-of-use and versatility of the technology.

While promising, existing CRISPR-Dx for COVID-19 have not considered the possibility that the viral sequences may be altered over time or in human cells. Viruses often mutate themselves especially under selective pressure. Thousands of SARS-CoV-2 genomes have been sequenced and deposited in the GISAID database[41,42] and analysis of their sequences has revealed many mutations, suggesting an ongoing adaptation of the coronavirus to its novel human host[43,44]. As a case in point, a new SARS-CoV-2 variant, termed VUI–202012/01, has recently emerged in the United Kingdom and is spreading very quickly throughout the region, leading to widespread lockdown in major cities like London. Importantly, researchers have also discovered mutations in the target sites of several qRT-PCR tests for COVID-19, which can affect the performance of these tests[45]. Similarly, mutations in the viral genome may also create mismatches in the guide RNA (gRNA) binding site and affect the Cas ribonucleoprotein (RNP) complex's ability to recognize its target. Furthermore, ADAR and APOBEC deaminases form part of the human host's innate immune responses to infection and have been shown to edit SARS-CoV-2 RNA[46,47]. The respective adenosine-to-inosine and cytosine-to-uracil changes may likewise prevent the CRISPR-Cas system from detecting the virus.

Besides a lack of robustness to variations in the SARS-CoV-2 RNA, there are other shortcomings of existing CRISPR-based assays. First, the duration of reported tests is generally around 40 min or so. In point-of-need scenarios, the waiting time should ideally be as short as possible. Hence, it is desirable if the CRISPR reaction can be sped up. Second, to boost sensitivity, CRISPR-Cas detection is typically combined with an isothermal amplification step, of which there are several options. Due to supply chain issues in the ongoing pandemic, reverse transcription loop-mediated isothermal amplification (RT-LAMP)[8] is the method-of-choice for COVID-19 applications. However, with the exception of AapCas12b and the TtCsm complex, the operating temperature for most Cas enzymes used in diagnostics is narrowly centred around 37 °C, while the RT-LAMP reaction is performed at 60–65 °C. Consequently, two heat blocks are required for many CRISPR diagnostics and time is also wasted in cooling the sample tubes. It is currently unclear if there are additional Cas enzymes that will allow the entire workflow to be performed at a single temperature. Third, most reported tests have only been evaluated on purified RNA samples. Consequently, it is unclear how well they will work on unpurified clinical specimens. Moreover, the process of RNA isolation adds at least 15 min to the test duration. Fourth, most reported tests do not have a built-in human internal control, which is essential for confirming that a negative result is not due to an insufficient amount of patient material. In the DETECTR system[17], separate reaction tubes are utilized for the human control and the actual SARS-CoV-2 test, but this setup is not ideal since one has to infer that the tube for the viral test contains the correct amount of sample input.

Here, we report the development of a CRISPR-based diagnostic assay for COVID-19 that addresses the above issues of existing tests. First, we incorporate design features that mitigate the loss in signal caused by viral genome mutations or RNA editing. In particular, we find that the use of enAsCas12a, an engineered E174R/S542R/K548R variant of AsCas12a[48], together with two gRNAs enhances the output signal when a variant nucleotide is present in the target substrate. Notably, while our assay can tolerate single nucleotide variations (SNVs) in the target sites, it still maintains high specificity and is able to distinguish SARS-CoV-2 from other coronaviruses reliably. Second, we discover that the use of modified guides improves reaction kinetics. Hybrid DNA-RNA guides work particularly well at our selected sites, increasing the on-target signal significantly compared to regular gRNAs while suppressing off-target background to negligible levels. Third, we discover that enAsCas12a exhibits an unexpectedly wide range of operating temperatures and is active from 37 to 65 °C. This property allows us to perform the entire RT-LAMP-CRISPR workflow in a single heat block. Fourth, we demonstrate how our assay can be applied on nasopharyngeal (NP) specimens directly without an extra RNA purification step, thereby improving the ease-of-use of our test. Fifth, we incorporate a human internal control into the same reaction tube, thereby simplifying the workflow even further. Taken together, our VaNGuard (Variant Nucleotide Guard) test holds the potential to address the need for a robust and rapid diagnostic assay that will help arrest viral spread and enable worldwide economies to re-open safely amidst the COVID-19 outbreak. Importantly, the various strategies presented here may also be adapted for use in future pandemics.

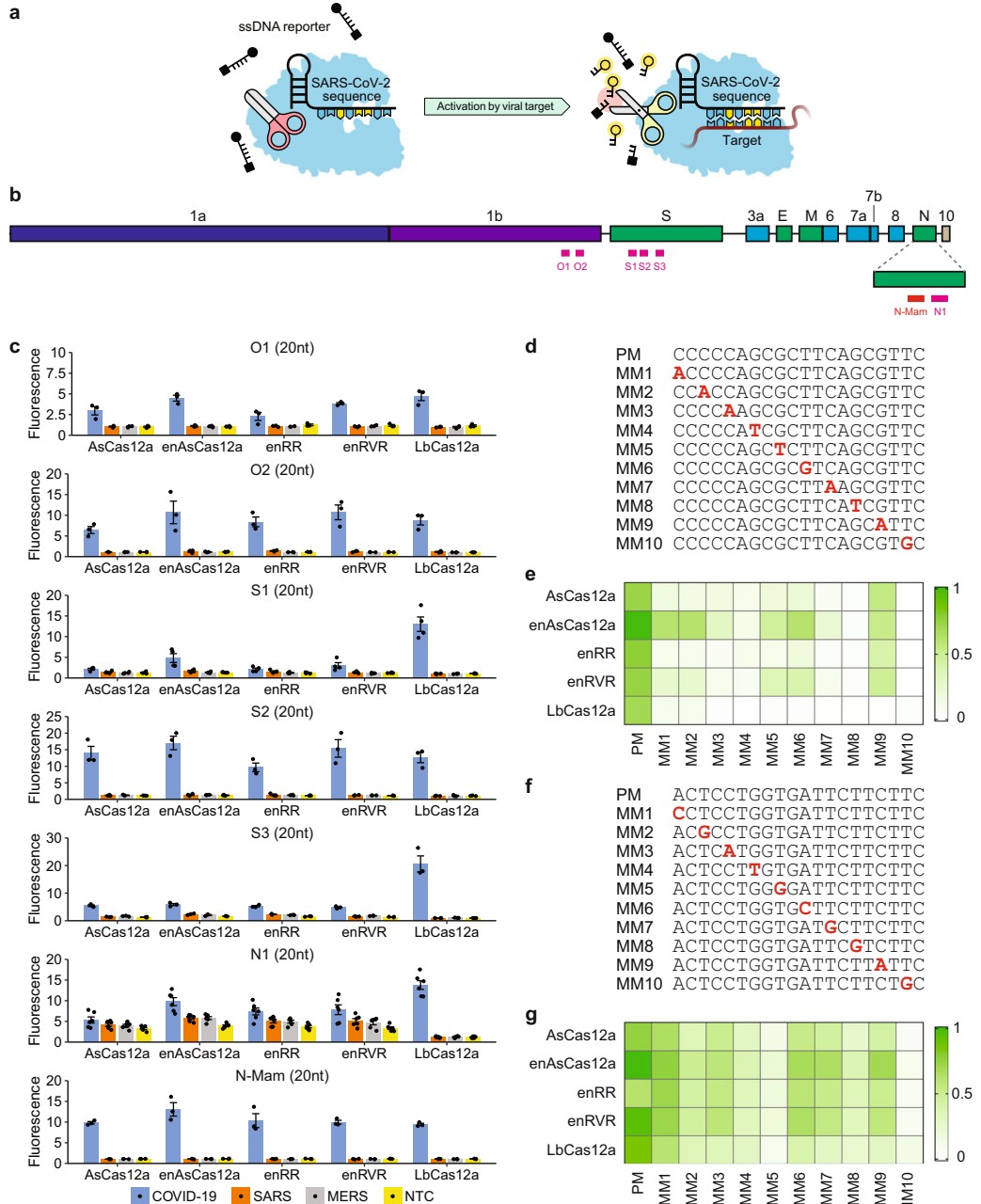

**Fig. 1 Evaluation of different Cas12a-gRNA combinations at room temperature (24 °C). a** Schematic of a fluorescence *trans*-cleavage assay. Here, the reporter comprises a fluorophore linked to a quencher by a short piece of ssDNA. The gRNA is programmed to recognize a particular locus of the SARS-CoV-2 genome. In the absence of the virus, the reporter molecule is intact and thus no fluorescence is observed. However, when the virus is present, the Cas12a RNP will bind to and cleave its programmed target, become hyperactivated, and cut the linker between the fluorophore and quencher, thereby generating a fluorescence signal. **b** Organization of the SARS-CoV-2 genome. Genes encoding structural proteins are indicated by green boxes, while genes encoding accessory proteins are indicated by cyan boxes. Although ORF10 is annotated in the genome, there is currently no evidence of its expression[71]. The locations of the new gRNAs are shown by pink bars below the genes, while the N-Mam locus is shown by a red bar. **c** Fluorescence measurements using a microplate reader after 30 min of cleavage reaction. 1E11 copies of the relevant DNA target were present in a 50 μl reaction. All readings were normalized to the no template control (NTC) at the start of the experiment. The N1 gRNA gave an unexpected result, whereby it triggered the collateral activity of AsCas12a and its variants without a template. Data represent mean ± s.e.m. (*n* = 3 [N-Mam, O1, O2, S2, S3], 4 [S1], or 6 [N1] biological replicates). **d** Sequences of perfect matched (PM) or mismatched (MM) spacers targeting the N-Mam locus. Each mismatched position is indicated by a bold red letter. **e** Heatmap showing the tolerance of various Cas12a enzymes to mismatched N-Mam gRNAs. The fluorescence readings are scaled between 0 and 1, where 1 is the highest measurement obtained and 0 is the background signal for NTC at the start of the experiment. **f** Sequences of perfect matched (PM) or mismatched (MM) spacers targeting the S2 locus. Each mismatched position is indicated by a bold red letter. **g** Heatmap showing the tolerance of various Cas12a enzymes to mismatched S2 gRNAs. The fluorescence readings are scaled between 0 and 1, where 1 is the highest measurement obtained and 0 is the background signal for NTC at the start of the experiment. Source data are available in the Source Data file.

## Results

**Characterization of Cas12a enzymes with different gRNAs.** We sought to evaluate the performance of various Cas12a RNP complexes in a fluorescence *trans*-cleavage assay (Fig. 1a) and benchmark them against that deployed in the DETECTR system[17], where wild-type LbCas12a was paired with a 20-nucleotide (nt) gRNA targeting the N-gene of SARS-CoV-2 (herein termed N-Mam gRNA) (Fig. 1b). To design new gRNAs, we aligned the genomes of SARS-CoV-2 and other related coronaviruses and selected six additional target sites (O1, O2, S1, S2, S3, and N1) that not only contained the TTTV protospacer adjacent motif (PAM) for Cas12a but were also highly divergent between the coronaviruses (Supplementary Fig. 1). We also purified five different Cas12a enzymes for testing. To assess the feasibility of a CRISPR-based diagnostic assay being deployed in a non-laboratory setting (e.g. a home setting), we initially carried out the cleavage reactions at room temperature using synthetic DNA fragments. Fluorescence was monitored over the course of 30 min in a microplate reader (Fig. 1c and Supplementary Fig. 2-4). For the N-Mam gRNA, we observed that while LbCas12a could detect SARS-CoV-2 with minimal cross-reactivity for SARS-CoV or MERS-CoV as expected, the other four Cas12a enzymes performed similarly, with enAsCas12a yielding an even higher fluorescence signal than LbCas12a in the presence of the SARS-CoV-2 substrate. The N-Mam gRNA was also not the most ideal for LbCas12a. The collateral activity of LbCas12a complexed with the S3 gRNA was approximately double that of the same enzyme complexed with the N-Mam gRNA in the presence of SARS-CoV-2. The S2 gRNA also generated stronger fluorescence signals than the N-Mam gRNA when paired with LbCas12a as well as with AsCas12a, enAsCas12a, and enRVR. Among all the tested enzymes, enAsCas12a exhibited the highest collateral activity with the S2 gRNA in the presence of SARS-CoV-2. Overall, the minimum spacer length for a gRNA in our diagnostic assay appeared to be 20nt. When we shortened the spacer length for either the N-Mam or the S2 gRNA to 18nt or 19nt, the collateral activity of all the Cas12a nucleases was reduced.

Next, we tested how mismatches at the gRNA-substrate interface may affect the fluorescence signal. We generated ten new gRNAs targeting the N-Mam locus, with each harbouring a single point mutation at variable locations along the spacer (Fig. 1d). From the *trans*-cleavage assay, we found that LbCas12a was very sensitive to imperfect base pairing between the gRNA and the target substrate, as any mismatch along the spacer reduced the fluorescence output to near-background levels, while AsCas12a and its variants were able to tolerate some of the mismatches (Fig. 1e and Supplementary Fig. 5). In particular, enAsCas12a was most tolerant of single nucleotide mismatches among the five tested enzymes. To verify the results, we generated ten additional gRNAs targeting the S2 locus, with each harbouring a point mutation at different positions along the spacer (Fig. 1f). Interestingly, we found that individual mismatches at the S2 locus affected the collateral activity of all the Cas12a endonucleases much less than those at the N-Mam locus (Fig. 1g and Supplementary Fig. 6). Nevertheless, enAsCas12a again exhibited the highest tolerance for point mutations, while wild-type LbCas12a was again the most sensitive to imperfect base pairing between the gRNA and its target substrate. We further confirmed the poor mismatch tolerance of LbCas12a by generating more mismatched (MM) gRNAs targeting the S3 locus and finding that the collateral activity of LbCas12a was greatly diminished for all the new MM gRNAs (Supplementary Fig. 7).

**Further characterization of enAsCas12a with S-gene gRNAs.** So far, we had performed the CRISPR-Cas detection at room temperature (24 °C) to simulate a non-laboratory setting, but we wondered if our diagnostic assay would perform substantially better at a more optimal reaction temperature (37 °C) and also if our observation of enAsCas12a being a more robust enzyme would still hold true at the higher temperature. Hence, we repeated the S2-targeting experiments at 37 °C. With the perfect matched (PM) gRNA, we observed that the fluorescence signal in our *trans*-cleavage assay increased around twice as fast at 37 °C in the presence of the intended SARS-CoV-2 template for all tested enzymes and reached much higher levels after 30 min of reaction, while showing little cross-reactivity for SARS-CoV and MERS-CoV (Fig. 2a and Supplementary Fig. 8). Furthermore, the activity profile in the presence of different point mutations remained similar, with enAsCas12a exhibiting the best mismatch tolerance as before (Fig. 2b and Supplementary Fig. 8). Hence, our results indicate that our CRISPR-based assay should be performed at 37 °C if a faster test result is desired and that enAsCas12a is a more suitable enzyme to use in a diagnostic test that is robust to viral genome mutations and intracellular RNA editing.

Although enAsCas12a exhibited higher mismatch tolerance than the other tested nucleases, its activity could still be appreciably affected by mismatches at certain positions along the gRNA-target interface (such as MM10). Hence, to further enhance robustness of the assay against variant nucleotides, we sought to combine two or more gRNAs with this enzyme. The S2 gRNA worked well with enAsCas12a, but the engineered nuclease showed poor *trans*-cleavage activities with both the S1 and S3 gRNAs. Hence, we screened additional guides targeting the region surrounding the S2 locus, so that when we coupled the CRISPR detection module with an isothermal amplification step, only one set of primers would be required. Based on genome sequences, each of the newly designed gRNAs was highly unique to SARS-CoV-2 (Supplementary Fig. 9) and covered over 99.5% of the isolates annotated in GISAID with no mismatches and insertions or deletions (indels) (Supplementary Data 2). From a fluorescence *trans*-cleavage assay, the S6 gRNA emerged as the most promising candidate because it exhibited the highest on-target activity for SARS-CoV-2 with little cross-reactivity for SARS-CoV and MERS-CoV (Fig. 2c and Supplementary Fig. 10).

Subsequently, we evaluated whether the newly identified S6 gRNA could rescue a mismatch at the S2 locus. To this end, we assembled the enAsCas12a nuclease with both the S6 gRNA and either a perfect matched (PM) or a mismatched (MM10) S2 gRNA. From a fluorescence *trans*-cleavage assay with synthetic DNA as substrate, we found that there was no significant difference in collateral activity between S2 PM gRNA and S2 MM10 gRNA when the S6 gRNA was present ($P > 0.2$, two-sided Student's *t*-test) (Fig. 2d and Supplementary Fig. 11). Furthermore, introduction of glycine, which was reported to improve the one-pot STOPCovid test[22], into the reaction also did not affect the Cas detection module significantly.

Next, we sought to combine RT-LAMP with our two-gRNA CRISPR detection module and compare the assay sensitivity in the absence or presence of a mismatch at the S2 locus. We tested three sets of LAMP primers and found one set that amplified well even with low amounts of input (Supplementary Fig. 12). With this selected primer set, we carried out RT-LAMP on variable copies of synthetic in vitro-transcribed (IVT) SARS-CoV-2 RNA templates at 65°C for 15 min before using the amplified products immediately in our *trans*-cleavage assay. Overall, we did not detect an obvious difference in sensitivity between the S2 PM gRNA and the S2 MM10 gRNA when the S6 gRNA was simultaneously deployed (Fig. 2e and Supplementary Fig. 13). Taken together, our results demonstrate that the use of two gRNAs can increase the robustness of CRISPR-Dx with respect to the presence of variant nucleotides.

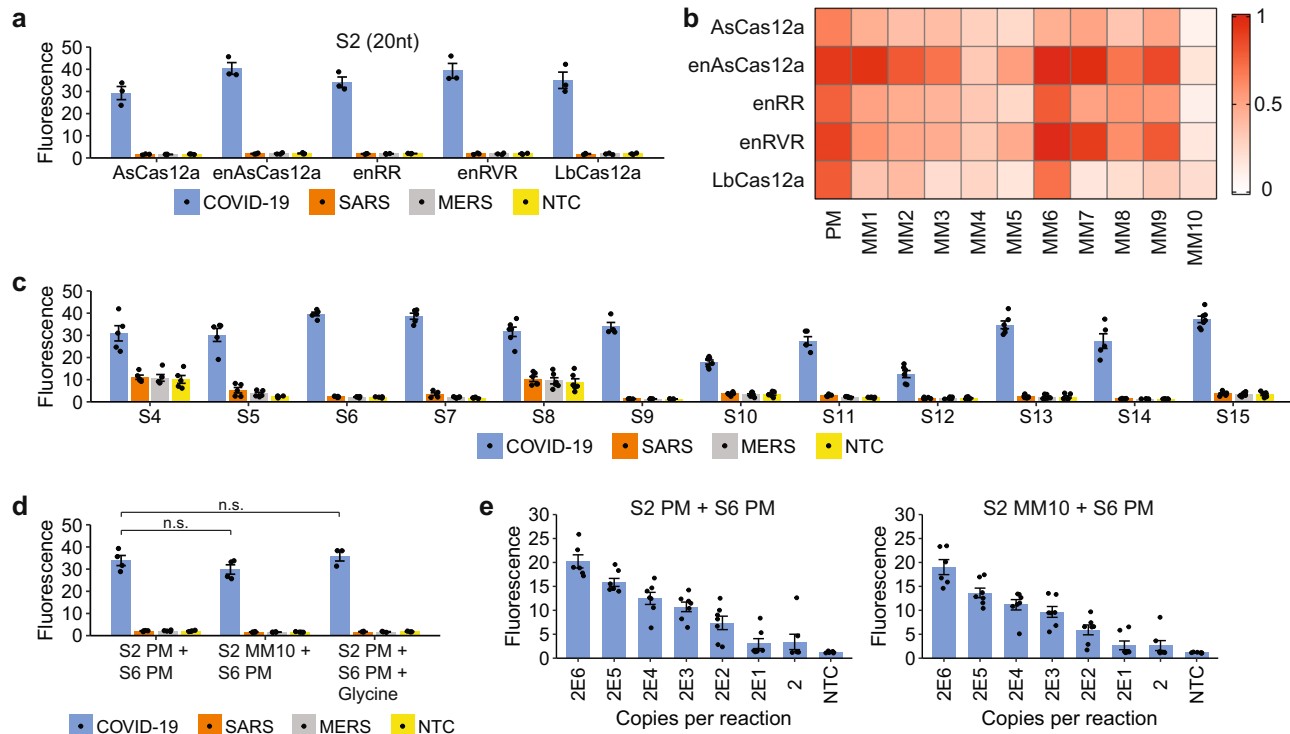

**Fig. 2 Activity and mismatch tolerance of enAsCas12a with various S-gene-targeting gRNAs. a** Fluorescence measurements for a single S2 gRNA complexed with various Cas12a nucleases after 30 min of *trans*-cleavage reaction at 37 °C. Compared to the earlier results obtained at 24 °C, there was still no cross-reactivity for SARS-CoV or MERS-CoV at the higher temperature, but the fluorescence signal for SARS-CoV-2 was approximately twice as high. Data represent mean ± s.e.m. ($n = 3$ biological replicates). **b** Heatmap showing the tolerance of various Cas12a enzymes to mismatches at the S2 target site when the *trans*-cleavage assay was performed at 37 °C. The fluorescence readings are scaled between 0 and 1, where 1 is the highest measurement obtained and 0 is the background signal for NTC at the start of the experiment. **c** Fluorescence measurements for enAsCas12a complexed with different gRNAs targeting the S-gene of SARS-CoV-2 after 30 min of cleavage reaction at 37 °C. 1E11 copies of DNA were present in a 50 µl reaction. Two of the gRNAs (S4 and S8) triggered the collateral activity of enAsCas12a without a template. Data represent mean ± s.e.m. ($n = 4$ [S9], 5 [S4, S5, S6, S7, S11, S14], 6 [S8, S10, S12, S13], or 7 [S15] biological replicates). **d** Buffering the collateral activity of enAsCas12a against SNVs with a second gRNA. Fluorescence measurements here were taken after 30 min of cleavage reaction at 37 °C. The S6 gRNA was used together with either the perfect matched (PM) or a mismatched (MM10) S2 gRNA in the absence or presence of 0.1 M glycine. Data represent mean ± s.e.m. ($n = 3$ [with glycine] or 4 [no glycine] biological replicates). (n.s. not significant, $P > 0.2$; two-sided Student's *t*-test). **e** Analytical limit of detection (LoD) for enAsCas12a complexed with both the S6 gRNA and either the perfect matched (PM) or a mismatched (MM10) S2 gRNA. Different copies of SARS-CoV-2 RNA fragments were used as input to RT-LAMP, which was performed at 65 °C for 15 min using an initial set of LAMP primers (0.2 µM of each displacement primer, 1.6 µM of each internal primer, and 0.8 µM of each loop primer). The Cas detection reaction was then carried out at 37 °C, with the fluorescence measurements here taken after 10 min. Data represent mean ± s.e.m. ($n = 6$ [2E6] or 7 [other copy numbers] biological replicates). Source data are available in the Source Data file.

**Buffering the LAMP reaction against variant nucleotides.** Besides the Cas detection module, mutations in or editing of the viral genome may also affect the isothermal amplification step. Hence, we sought to determine which LAMP primers were more susceptible to mismatches at their binding sites. The original method is based on two internal primers, known as FIP and BIP, and two displacement primers, known as F3 and B3, which collectively target six distinct regions in the DNA template (Fig. 3a). We hypothesized that mismatches at the 3′ end of each primer may affect extension by the Bst DNA polymerase. Therefore, we tested the RT-LAMP reaction with either perfect matched (PM) primers or primers with a mismatch (MM) positioned at the first, second, or third nucleotide from the 3′end (Fig. 3b). Moreover, since imperfect base pairing may also affect extension from the free 3′ end of the dumbbell DNA generated during LAMP (Fig. 3a), we further tested the reaction with FIP or BIP primers carrying a mismatch at their 5′ ends too (Fig. 3b). The RT-LAMP reaction was monitored in real-time with a fluorescent dye. Our experiments revealed that mismatches in the two displacement primers did not affect the amplification step appreciably (Fig. 3c). In contrast, mismatches at the 3′ ends of FIP and BIP as well as at

the 5′ end of FIP reduced the rate of amplification significantly ($P < 0.05$, one-sided Student's *t*-test).

Next, we sought to develop or apply strategies to enhance the robustness of the LAMP reaction against potential variant nucleotides. We reasoned that since MM1 caused the greatest reduction in amplification efficiency, the use of a FIP or BIP primer that was truncated right at the end would avoid the most harmful mismatch altogether (Fig. 3b). Indeed, usage of a mixture of the original internal primers and the truncated primers (tPM-3 or tPM-5) led to a significant improvement in the amplification rate ($P < 0.05$, one-sided Student's *t*-test) (Fig. 3d, e). Furthermore, we noted that the Bst DNA polymerase lacked a 3′-to-5′ exonuclease activity and thus would encounter difficulty extending any DNA with mismatches at the 3′ end. One possible solution was to add to the LAMP reaction a small amount of high-fidelity DNA polymerase, which possessed a proofreading capability and thus may help to remove any mismatched bases at the 3′ end[49]. Indeed, we found that addition of 0.15U high-fidelity polymerase did result in a significant increase in the rate of reaction despite the presence of end mismatches in FIP or BIP ($P < 0.05$, one-sided Student's *t*-test) (Fig. 3d, e). Notably, in the

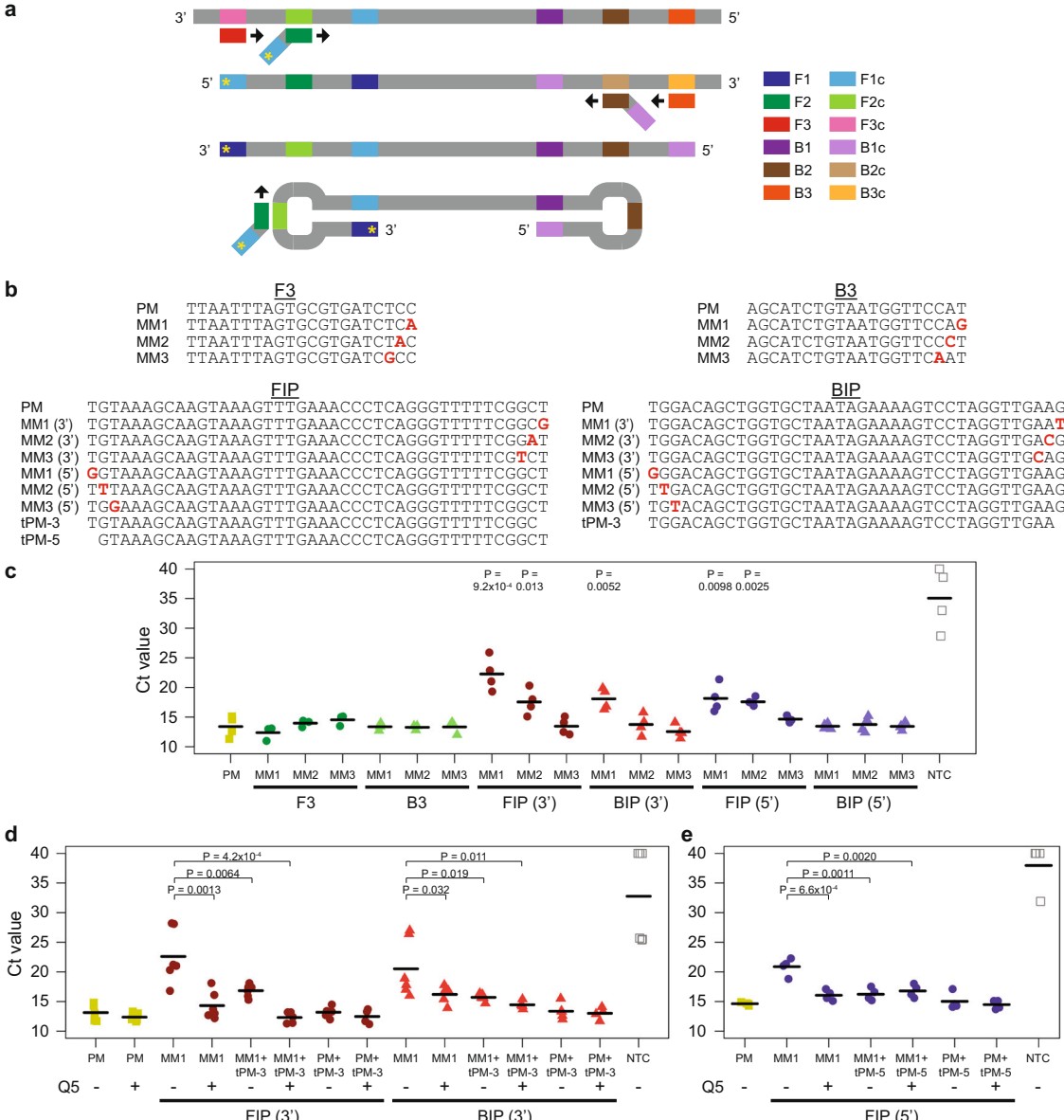

**Fig. 3 Evaluating and enhancing the robustness of LAMP. a** Schematic of LAMP. Six distinct regions in the target locus (F1, F2, F3, B1, B2, and B3) are recognized by four core primers, which have a black arrow at their 3′ ends to represent extension by the DNA polymerase. FIP is represented by a dark green rectangle joined to a slanted light blue rectangle. BIP is represented by a dark brown rectangle joined to a slanted light purple rectangle. In addition, the F3 displacement primer is represented by a red rectangle, while the B3 displacement primer is represented by an orange rectangle. The letter "c" appended to each region name indicates the reverse complementary sequence. After our LAMP optimization process, we incorporated swarm primers, whose sequences are equivalent to F1c and B1c. Moreover, to demonstrate how a mismatch at the 5′ end of FIP can affect the LAMP reaction, we have added a yellow asterisk to track the progression of the mismatch. **b** Sequences of LAMP primers tested. The mismatches are indicated by bold red letters. **c** Strip chart showing how mismatches between LAMP primers and their binding sites affected the rate of isothermal amplification. RT-LAMP was performed at 65 °C in a real-time instrument with 20,000 copies of synthetic RNA corresponding to the S-gene of SARS-CoV-2. Cycle-threshold (Ct) values were given by the instrument using default settings. The black horizontal bars among the data points in the strip chart represent the mean ($n = 3$ [F3 MM, B3 MM] or 4 [PM, FIP (3′) MM, BIP (3′) MM, FIP (5′) MM, BIP (5′) MM, NTC] biological replicates). P-values were calculated using one-sided Student's t-test. **d** Strip chart showing rescue of the LAMP reaction by truncated primers and a Q5 high-fidelity DNA polymerase in the presence of mismatches at the 3′ ends of FIP and BIP. RT-LAMP was performed at 65 °C with 20,000 copies of RNA template. The black horizontal bars among the data points in the strip chart represent the mean ($n = 4$ [FIP PM + tPM+Q5, BIP PM + tPM+Q5], 5 [PM, PM + Q5, FIP MM + tPM+Q5, FIP PM + tPM, BIP MM + tPM+Q5, BIP PM + tPM], or 6 [FIP MM, FIP MM + Q5, FIP MM + tPM, BIP MM, BIP MM + Q5, BIP MM + tPM, NTC] biological replicates). P-values were calculated using one-sided Student's t-test. **e** Strip chart showing rescue of the LAMP reaction by truncated primers and a Q5 high-fidelity DNA polymerase in the presence of a mismatch at the 5′ end of FIP. RT-LAMP was performed at 65 °C with 20,000 copies of RNA template. The black horizontal bars among the data points in the strip chart represent the mean ($n = 4$ biological replicates). P-values were calculated using one-sided Student's t-test. Source data are available in the Source Data file.

case of mismatches at the 3' ends of the internal primers, an even larger improvement in amplification efficiency was observed when the two strategies of truncated primers and high-fidelity DNA polymerase were used together (Fig. 3d). Collectively, our results indicate that we can enhance the robustness of the LAMP reaction against variant nucleotides by utilizing two sets of internal primers (full-length and tPM-3) and a high-fidelity polymerase.

**Strategies to enhance the sensitivity of LAMP.** An important metric used to assess the performance of a diagnostic assay is its sensitivity. Although our assay contained features to handle SNVs in the viral genome, we noted that as the copy number decreased from 2E6 to 2, the test sensitivity declined monotonically as well (Fig. 2e and Supplementary Fig. 13). Hence, we sought to improve the sensitivity of our assay.

First, we tested how variations in the primer concentrations might affect the LAMP reaction. We focused on the displacement primers (F3 and B3) and the internal primers (FIP and BIP), which were part of the original LAMP setup. 20 copies of RNA template were used as input and the reaction was monitored in a real-time instrument with a fluorescent dye (Fig. 4a). Overall, we observed that doubling the concentration of F3, FIP, or BIP worsened the performance of our CRISPR-Dx. In contrast, when we increased the amount of B3 by twofold, the assay sensitivity improved marginally with 75% (9 out of 12) of the replicates showing successful amplification. This might be because the B3 primer given by the PrimerExplorer design software (https://primerexplorer.jp/e/) was sub-optimal.

Enzyme engineering may improve the performance of LAMP. Hence, second, we compared several different commercially available Bst polymerases, namely Bst2.0, Bst3.0 (a mutant polymerase with an intrinsic reverse transcriptase activity), and Turbo Bst (a polymerase fused to an extra DNA-binding domain), using our RT-LAMP setup (Supplementary Fig. 14). While Bst3.0 alone performed worse than the original Bst2.0 mastermix that we had been using, Bst3.0 with a separate reverse transcriptase added significantly enhanced the kinetics of the reaction ($P < 0.01$, one-sided Student's $t$-test). However, we observed that the combination of Bst3.0 and the additional reverse transcriptase was highly prone to false amplification, giving a fluorescence signal even in the absence of template. Moreover, we found that while Turbo Bst also improved the reaction kinetics marginally, it too was significantly more prone to false amplification than Bst2.0 ($P < 0.01$, one-sided Student's $t$-test). Therefore, we retained the use of Bst2.0 mastermix for subsequent experiments.

Third, we asked if the use of chemical additives might bolster the sensitivity of LAMP. A recent study showed that glycine and taurine could improve the kinetics of the one-pot STOPCovid test[22]. However, it is unclear if the improvement occurs in the RT-LAMP reaction or in the Cas detection module. To address the question, we performed the RT-LAMP reaction only with or without either of the two chemicals. Overall, our data revealed that addition of glycine enhanced the sensitivity of RT-LAMP, with over 90% (11 out of 12) of the replicates showing successful amplification of 20 copies of viral template (Fig. 4b). Addition of taurine also improved assay sensitivity in a similar manner to glycine, although usage of both chemicals together did not have a synergistic effect (Supplementary Fig. 15). Besides glycine and taurine, another study reported that dimethyl sulfoxide (DMSO) increased the sensitivity and specificity of LAMP reactions[50]. The organosulfur compound is also often used in PCR to disrupt secondary structures of GC-rich templates. However, we found that both 2.5% and 5% DMSO exerted an inhibitory effect on

LAMP instead (Supplementary Fig. 16). Moving forward, we incorporated glycine into our assay, since it is commonly found in laboratories.

Fourth, we wondered if alternative LAMP reaction schemes would deliver higher sensitivities. Although the earliest LAMP method relied on four core primers[8], subsequent studies described improvements in the method due to the addition of new primer sets. The most commonly added primer set is the "loop primers" (LF and LB), which target the single-stranded loop regions in the dumbbell structures generated during the reaction[51]. The loop primers are provided with the four core primers by the PrimerExplorer design software. Furthermore, two other primer sets that may be added include the "stem primers", which target the single-stranded region between F1/F2/F3 and B1/B2/B3 (Fig. 3a)[52], and the "swarm primers", which anneal to the template strand opposite to that of FIP or BIP so as to expose the binding sites for the internal primers[53]. Therefore, we tested new stem primers and swarm primers in conjunction with our previous set of LAMP primers targeting the S-gene of SARS-CoV-2. The data from our initial set of experiments indicated that although addition of stem primers (Stem$_{in}$) were detrimental to the RT-LAMP reaction possibly due to the short region available between F1/F2/F3 and B1/B2/B3, addition of swarm primers improved the reaction kinetics significantly ($P < 0.05$, one-sided Student's $t$-test) (Fig. 4c). Notably, this improvement was only observed when the swarm primers were used with the core primers and the loop primers. When the loop primers were omitted, amplification occurred much later than our original RT-LAMP setup, indicating that the swarm primers could not substitute for the loop primers. Next, we examined the design of our stem primers (Stem$_{in}$) and noticed that they pointed towards each other with their 3' ends competing for binding to the template. Hence, we tested each of the primers individually and also evaluated another pair of stem primers (Stem$_{out}$) that pointed away from each other. Our results showed that one Stem$_{in}$ primer alone as well as the Stem$_{out}$ primers were able to improve the kinetics of LAMP reaction to varying extents (Fig. 4d). Moreover, addition of one or two stem primers to the cocktail of core, loop, and swarm primers did not further improve the kinetics of LAMP (Supplementary Fig. 17). Therefore, moving forward, we did not continue to pursue the stem primers and focused mainly on the loop and swarm primers. Our final set of LAMP primers is highly specific to SARS-CoV-2 as shown by the sequence alignment of multiple coronaviral genomes (Supplementary Fig. 18).

We assessed if the optimized LAMP conditions together with our two-gRNA (S2 and S6) CRISPR detection module could improve assay sensitivity and accommodate point mutations. Unlike the original LAMP conditions without any swarm primers or glycine (Fig. 2e and Supplementary Fig. 13), we found that the fluorescence signal did not drop as much with decreasing amounts of RNA template when we utilized the optimized conditions (Fig. 4e and Supplementary Fig. 19a). The analytical limit of detection (LoD) was 20 copies per reaction with our optimized conditions, regardless of the absence or presence of a mismatch at the S2 locus. This sensitivity was further confirmed by a lateral flow assay (Supplementary Fig. 19b), which is a convenient paper-based platform to readout the results (Supplementary Fig. 20).

Instead of artificially creating mismatches in the gRNA, we sought to evaluate the robustness of our assay with a real-life mutation in the target template using PM gRNAs. To this end, we selected a known S254F mutation in the S-gene[54–57], which may potentially interfere with the binding of the S2 gRNA. Upon targeting of the mutant viral RNA with enAsCas12a complexed to the S2 gRNA alone, we observed low levels of fluorescence that are close to background for template amounts up to 2E6 copies

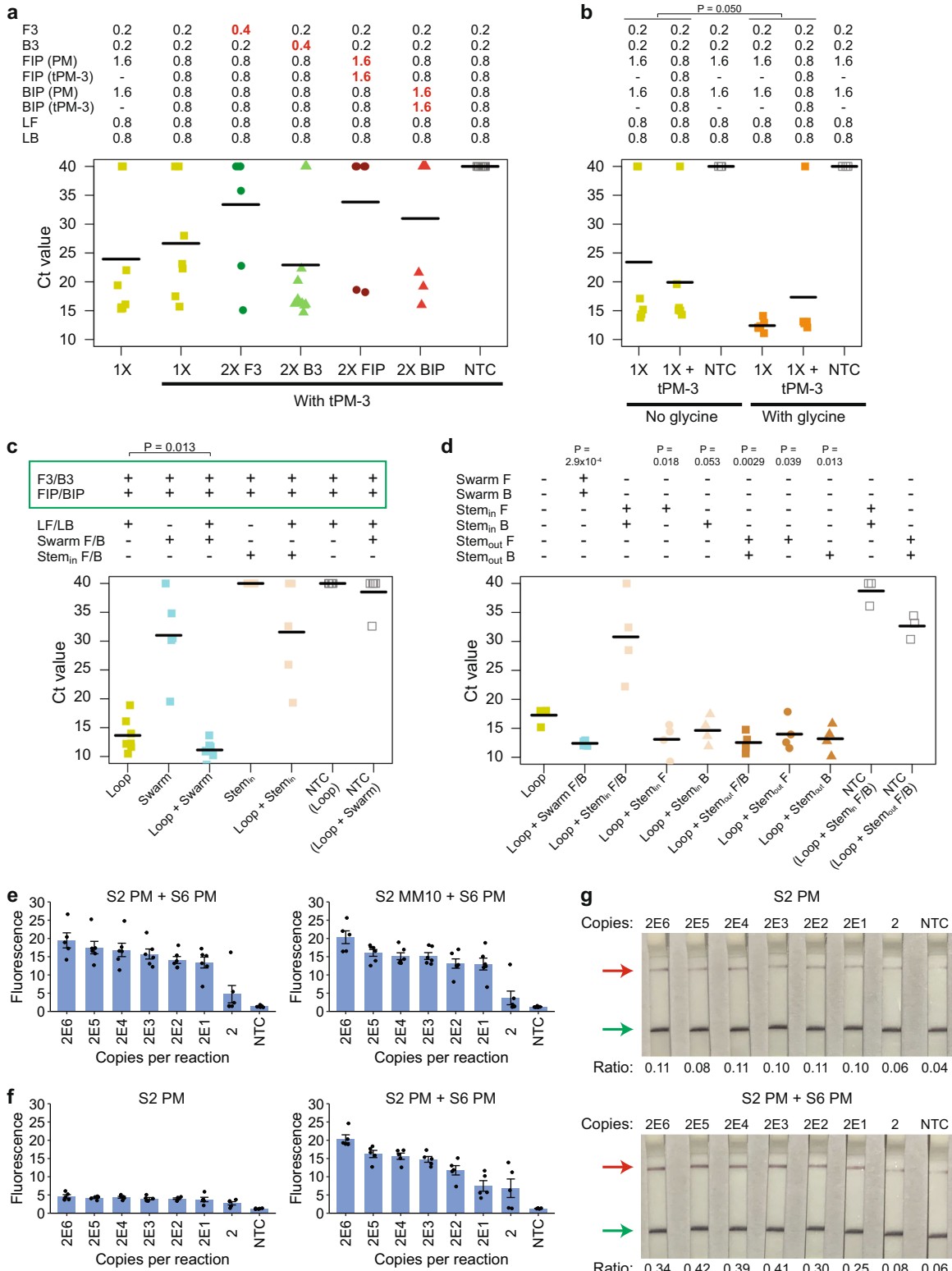

(Fig. 4f and Supplementary Fig. 21). In contrast, targeting of the S254F mutant template with both the S2 and S6 gRNAs yielded much higher fluorescence signals. We further confirmed these results with a lateral flow assay, where a positive test outcome was clearly obtained with two gRNAs even when only 20 copies of the mutant template were present (Fig. 4g). Collectively, the data indicate that our assay is robust against variant nucleotides in the viral target and can detect low copies of SARS-CoV-2.

**Detection of viral RNA in total human RNA samples.** Having evaluated the performance of our VaNGuard test with pure synthetic RNAs, we next sought to evaluate our assay in more realistic situations. Specifically, we wondered if the SARS-CoV-2 RNA could still be detected in a large pool of human RNA. First, we spiked in 20,000 copies of in vitro-transcribed viral RNA into 10 ng of total RNA extracted from various human cell lines and then performed RT-LAMP under optimized conditions followed

**Fig. 4 Methods to improve sensitivity of LAMP. a** Strip chart showing how LAMP sensitivity was affected by the concentration of primers used. We tested different concentrations of displacement primers and internal primers. RT-LAMP was performed at 65 °C in a real-time instrument with 20 copies of RNA template corresponding to the S-gene of SARS-CoV-2. The black horizontal bars among the data points in the strip chart represent the mean (n = 7 [1X, 2X F3, 2X FIP, 2X BIP], 10 [1X without tPM], 12 [2X B3], or 15 [NTC] biological replicates). **b** Strip chart showing how LAMP sensitivity was altered by 0.1 M glycine. RT-LAMP was performed at 65 °C with 20 copies of RNA template. The black horizontal bars among the data points in the strip chart represent the mean (n = 3 [NTC] or 6 [with template] biological replicates). P-value was calculated using one-sided Student's t-test. **c** Strip chart showing how LAMP sensitivity was altered by the use of swarm or stem primers. The green box demarcates the four core primers, which were included in every experiment. The concentrations of each displacement primer, internal primer, loop primer, swarm primer, and stem primer were 0.2, 1.6, 0.8, 1.6, and 1.6 μM, respectively. RT-LAMP was performed at 65 °C with 20,000 copies of RNA template. The black horizontal bars among the data points in the strip chart represent the mean (n = 5 [swarm, stem$_{in}$, loop + stem$_{in}$, NTC for loop + swarm] or 7 [loop, loop + swarm, NTC for loop] biological replicates). P-value was calculated using one-sided Student's t-test. **d** Further dissection of stem primers. The strip chart shows the impact of various stem primers on LAMP sensitivity. Here, every reaction contained the displacement primers (0.2 μM each), internal primers (1.6 μM each), and loop primers (0.8 μM each). Furthermore, it could also contain either two additional swarm primers or one or two additional stem primers (with the concentration of each extra primer being 1.6 μM). RT-LAMP was performed at 65 °C with 20,000 copies of RNA template. The black horizontal bars among the data points in the strip chart represent the mean (n = 3 [NTC] or 4 [with template] biological replicates). P-values were calculated using one-sided Student's t-test. **e** Analytical LoD for enAsCas12a complexed with both the S6 gRNA and either the PM or the MM10 S2 gRNA. RT-LAMP was performed at 65 °C for 15 minutes under optimized conditions, which encompassed doubling the concentration of B3 to 0.4 μM, using both full-length and 1nt-truncated internal primers (1.6 μM each), including the swarm primers (1.6 μM each), and adding 0.15U Q5 polymerase and 0.1 M glycine into each reaction. Fluorescence readings using a microplate reader after 10 min of cleavage reaction at 37 °C are shown. Data represent mean ± s.e.m. (n = 5 [2E6] or 6 [other copy numbers] biological replicates). **f** Analytical LoD for enAsCas12a when a S254F mutation was present in the viral template. The nuclease was assembled either with the S2 gRNA alone or with both the S2 and S6 gRNAs. These gRNAs were designed to be perfect matched against the reference SARS-CoV-2 genome. RT-LAMP was performed at 65 °C for 15 min under optimized conditions. Fluorescence readings after 10 min of cleavage reaction at 37 °C are shown. Data represent mean ± s.e.m. (n = 5 biological replicates). **g** Similar experiments to **f**, except that a different reporter was used and a dipstick was added to each sample tube after 10 minutes of cleavage reaction. Bands appeared on the dipsticks by 2 min. The red arrow indicates the test bands, while the green arrow indicates the control bands. Ratios of test band intensity to control band intensity are given under each dipstick. Source data are available in the Source Data file.

by our fluorescence *trans*-cleavage assay with both the S2 and S6 gRNAs (Fig. 5a and Supplementary Fig. 22). Our data revealed that the presence of a complex pool of human RNA did not significantly affect the fluorescence signal of the assay (P > 0.2, one-sided Student's t-test). We also tested if the presence of human RNA and genomic DNA together might interfere with the detection of the virus, but did not observe any appreciable loss of fluorescence signal either.

Second, we asked whether the presence of a complex pool of human RNA would affect the sensitivity of our assay for COVID-19 when the viral sequence had been mutated or edited. To this end, we generated a synthetic viral template harbouring not only the S254F mutation but also a second silent N234N mutation that had been found in at least ten sequenced SARS-CoV-2 isolates from around the world (Supplementary Data 2). While the former mutation could affect target recognition by the S2 gRNA, the latter mutation may affect target binding by the S6 gRNA. We examined the sensitivity of our assay using this double mutant viral template either by itself or in a pool of total human RNA from the HCC2279 lung cell line. Encouragingly, we found that the LoD remained at 20 copies per reaction in both cases (Fig. 5b and Supplementary Fig. 23), underscoring the robustness of our VaNGuard test against known mutations in the viral RNA even in the presence of total human RNA.

Third, since usage of patient samples directly without an extra RNA isolation step would reduce the time and cost of a diagnostic test, we examined whether various sample collection media may affect the performance of our assay (Supplementary Fig. 24). Strikingly, just 1 μl of a commercially available SAFER Sample reagent was sufficient to block the RT-LAMP reaction completely. In contrast, up to 4 μl of Universal Transport Medium (UTM) could be tolerated with the kinetics of RT-LAMP reduced only marginally. The isothermal amplification reaction could also accommodate up to 4 μl of QuickExtract, albeit to a lesser extent than UTM. Nevertheless, when we studied the impact of UTM on the entire RT-LAMP-CRISPR workflow, we observed that 4 μl of UTM clearly reduced the sensitivity of our VaNGuard test

compared to just 2 μl of this widely used collection medium (Fig. 5c and Supplementary Fig. 25). Hence, we conclude that up to 2 μl (or less than 10% volume) of UTM may be added into our assay without any adverse consequence on its performance. Subsequently, we examined the assay sensitivity using either wild-type or double mutant viral template mixed with total RNA from HCC2279 cells in 2 μl of UTM and found that the presence of two mutations in the gRNA binding sites still did not degrade the LoD of our assay (Fig. 5d and Supplementary Fig. 26). Altogether, our results suggest that patient samples in a small amount of UTM may be used directly in our VaNGuard test without affecting its robustness against SNVs in the template.

**Reaction conditions affecting enAsCas12a collateral activity.** In our earlier lateral flow assays, although positive test outcomes were obtained for samples with at least 20 copies of synthetic SARS-CoV-2 RNA, we noticed that the test bands were relatively weak compared to the control bands (Fig. 4g and Supplementary Fig. 19b). We hypothesized that we might be using a sub-optimal buffer (Buffer 3.1) for the enAsCas12a-mediated assay. Hence, we tested an alternative reaction buffer (Buffer 2.1) together with different test durations and higher concentrations of the Cas12a RNP (Fig. 6a). Overall, we observed that reactions in the original buffer exhibited slower kinetics than reactions in the alternative buffer. For example, the intensity of the test band after 20 min in Buffer 3.1 was achieved by around 10 min in Buffer 2.1. Moreover, increasing the concentration of the Cas12a RNP by at least 50% also boosted the test signal. Hence, we re-evaluated the sensitivity of our VaNGuard test using Buffer 2.1 and in vitro-transcribed SARS-CoV-2 RNA templates (Fig. 6b). Stronger test bands were observed from 2 to 2E6 copies of wild-type and S254F mutant viral templates with the Cas detection reaction performed for just 10 min.

Encouraged by these results, we systematically investigated the reaction conditions under which purified enAsCas12a protein was active in vitro. Specifically, we tested four distinct buffers

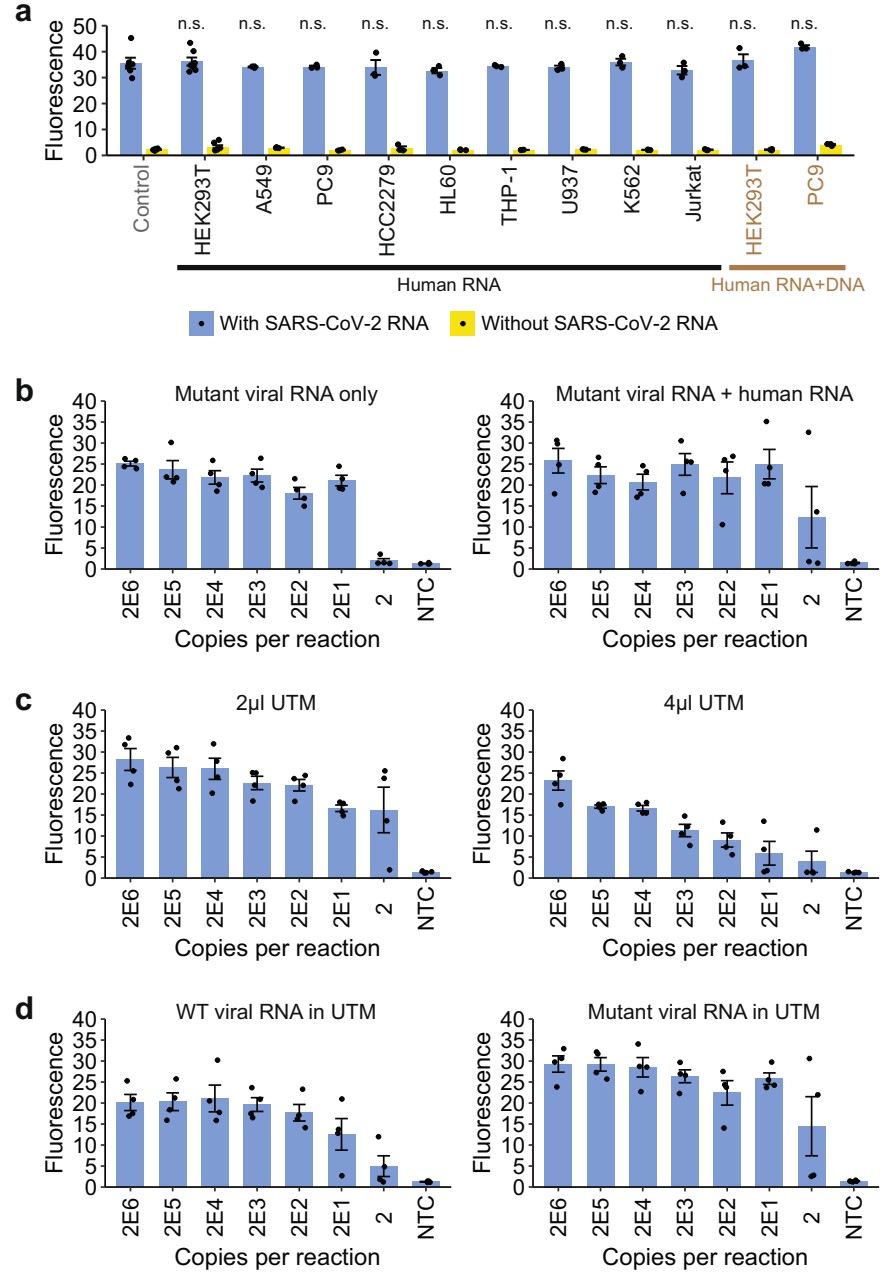

**Fig. 5 Evaluation of the VaNGuard test under more realistic conditions. a** 20,000 copies of synthetic SARS-CoV-2 RNA fragments were spiked into 10 ng of total RNA extracted from different immortalized human cell lines to mimic infection in various cell types [HEK293T: adrenal precursor, A549: lung, PC9: lung, HCC2279: lung, HL60: blood (promyelocytes), THP-1: blood (monocytes), U937: blood (monocytes), K562: blood (granulocytes/ erythrocytes), and Jurkat: blood (T cells)]. Pure synthetic viral RNAs were used as a control. The control or spiked RNA samples served as input to RT-LAMP, which was performed at 65 °C for 15 min. The Cas detection reaction was then carried out at 37 °C, with fluorescence after 30 min shown. Data represent mean ± s.e. m. (n = 3 [all except control and HEK293T RNA], 6 [control], or 7 [HEK293T RNA] biological replicates). There was no significant loss of signal in the presence of human RNA (n.s.: not significant, P > 0.2; one-sided Student's t-test). **b** Analytical LoD for a S254F N234N double mutant RNA template either by itself or mixed with 10 ng total human RNA from HCC2279 cells. Different copies of synthetic viral template (with or without human RNA) were used as input to RT-LAMP, which was performed at 65 °C for 15 min. The Cas detection reaction was then carried out at 37 °C, with fluorescence after 10 min shown. Data represent mean ± s.e.m. (n = 4 biological replicates). **c** Analytical LoD for purified synthetic wild-type SARS-CoV-2 RNA template in the presence of 2 μl or 4 μl UTM. Data represent mean ± s.e.m. (n = 4 biological replicates). **d** Analytical LoD for wild-type (WT) or S254F N234N double mutant RNA template mixed with 10 ng total human RNA from HCC2279 cells in the presence of 2 μl UTM. Data represent mean ± s.e.m. (n = 4 biological replicates). Source data are available in the Source Data file.

with or without dithiothreitol (DTT) over a range of temperatures using the S2 gRNA with enAsCas12a (Fig. 6c and Supplementary Fig. 27). Unexpectedly, we found that enAsCas12a was active in our *trans*-cleave assay at all the temperatures tested. Overall, the engineered enzyme performed better in buffers containing acetate salts (CutSmart and Tango) than in buffers containing chloride salts (Buffer 2.1 and Buffer 3.1). At 37 °C, CutSmart with DTT emerged as the best buffer to use with enAsCas12a, while at 60 °C, Tango was the most suitable. Addition of DTT helped certain reaction conditions, for example CutSmart at 37 °C

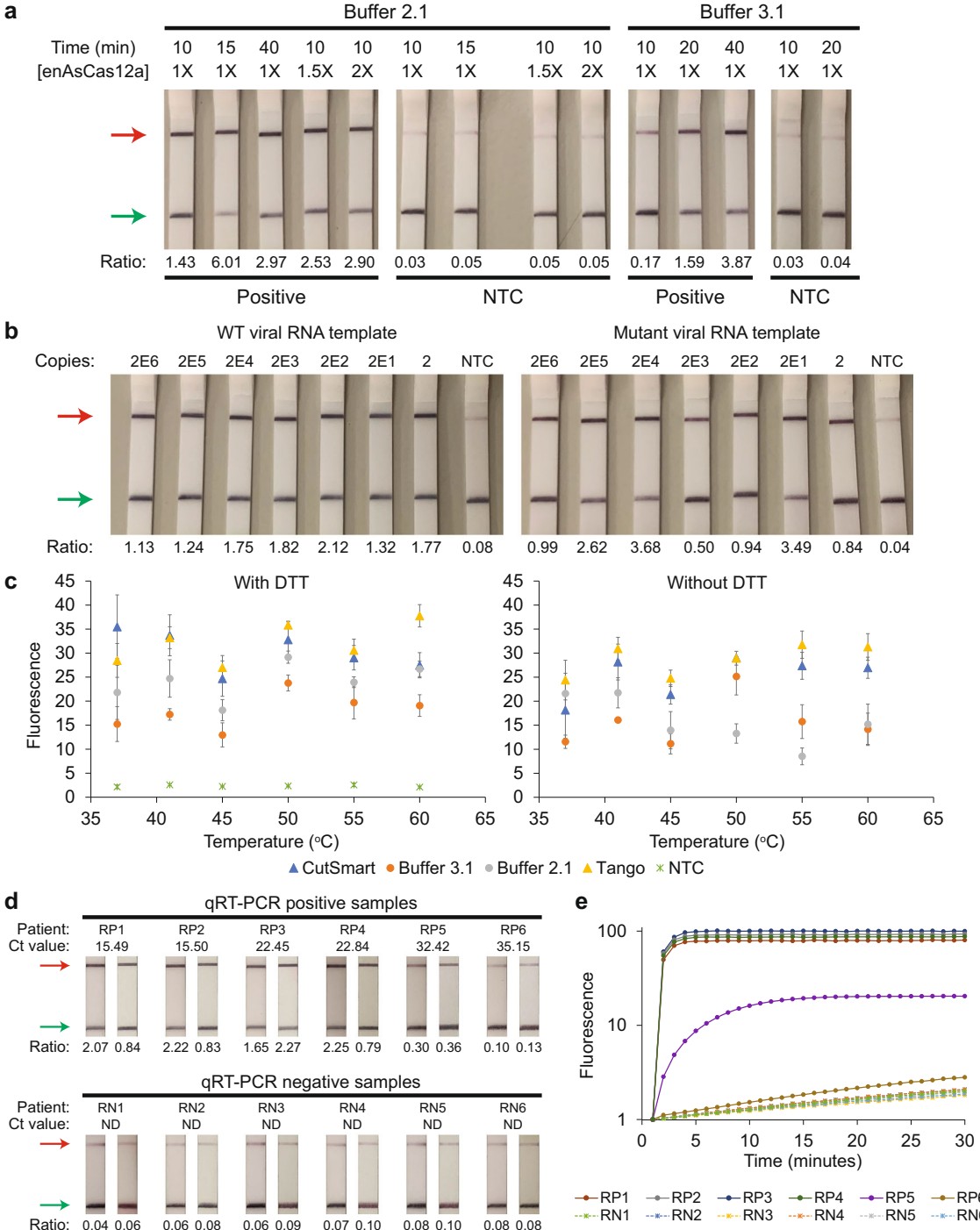

(Supplementary Fig. 28). We further confirmed that DTT was required in the CutSmart buffer for enAsCas12a to function optimally at 37 °C using the S6 gRNA (Supplementary Fig. 29).

Subsequently, we applied our diagnostic assay on a pilot set of leftover RNA samples extracted from patient nasopharyngeal (NP) swabs that had previously been analyzed by qRT-PCR in the hospital (Fig. 6d). We selected samples that exhibited a range of Ct values and performed the Cas detection step at 37 °C in CutSmart buffer with DTT. All the six samples that were negative in qRT-PCR analysis also turned out to be negative in the lateral flow assay, suggesting a specificity of 100% for our VaNGuard test. In addition, five out of the six infected samples gave obvious positive results on the dipsticks, with the remaining sample yielding a test band whose normalized intensity was only slightly

above that of background. We confirmed the results by repeating the test on the same set of patient samples using a fluorescence readout on the real-time instrument instead (Fig. 6e). Hence, our assay appeared to be able to detect SARS-CoV-2 in clinical RNA samples containing at least 93 copies of the virus, which corresponded to a cycle-threshold (Ct) value of 32.42 for the qRT-PCR kit used.

**Engineering of guides to enhance the sensitivity of Cas12a detection.** From the pilot evaluation with clinical samples, we observed that while our assay gave a positive test result for an infected sample with a Ct value of 32.42, the test band intensity and the fluorescence signal were weaker than those of samples

**Fig. 6 Optimizing reaction conditions for enAsCas12a. a** Evaluation of various experimental conditions, including different concentrations of enAsCas12a and different durations of the cleavage reaction. 1X specifies 65 nM. 2E6 copies of synthetic wild-type SARS-CoV-2 RNA served as input to RT-LAMP. **b** Detection of wild-type or S254F mutant SARS-CoV-2 sequence using S2 and S6 gRNAs. Different copies of SARS-CoV-2 RNA fragments were used as input to RT-LAMP, which was performed at 65 °C for 15 min. Next, the Cas detection reaction was carried out at 37 °C for 10 min in Buffer 2.1 with DTT before a dipstick was added to each reaction tube. **c** Systematic testing of different reaction buffers and temperatures for the Cas detection step. Here, enAsCas12a complexed with the S2 gRNA only was utilized in a 50 μl *trans*-cleavage assay with 2E11 copies of DNA template corresponding to SARS-CoV-2 S-gene. Data represent mean ± s.e.m. (n = 3 [41 °C, 45 °C, 50 °C, 55 °C Tango alone, 60 °C no DTT and NTC], 4 [37 °C, 55 °C all but Tango alone], or 6 [60 °C with DTT] biological replicates). **d** Preliminary evaluation of our VaNGuard test with leftover patient samples. A Ct value of 30 was estimated to be equivalent to 500 copies of the virus. RT-LAMP was performed at 65 °C for 15 min before the Cas detection reaction was carried out at 37 °C for 10 min in CutSmart with DTT. Each clinical sample was tested twice using dipsticks. **e** Retesting the pilot set of clinical RNA samples using a fluorescence readout. RT-LAMP was performed at 65 °C for 15 min. Subsequently, 4 μl LAMP products (out of 25 μl) were used for the *trans*-cleavage assay, which was performed at 37 °C in a real-time instrument where measurements were taken every minute. The fluorescence readings for all six clinically negative samples remained low over the duration of the experiment. Additionally, the fluorescence readings for five out of the six clinically positive samples showed a clear exponential increase with time. The remaining positive sample (RP6), which contained 14 copies of the virus, gave fluorescence signals that were only slightly above those of the negative samples. Source data are available in the Source Data file.

with higher viral loads (Fig. 6d, e). Hence, we asked if the sensitivity of the Cas detection module could be enhanced. To this end, we sought to determine if modified gRNAs would boost the in vitro cell-free cleavage activity of our purified Cas12a RNPs, since previous studies had reported that such gRNAs could increase the *cis*- and *trans*-cleavage activity of Cas9 and Cas12a nucleases[32,58–64]. First, we tested if extensions of the gRNA at its 3' end would enhance the activity of our Cas12a enzymes. We tried $U_3$, $U_8$, and $U_4AU_6$ extensions, but unlike previous work[32,62], we did not observe an appreciable or consistent improvement in activity (Supplementary Fig. 30).

Second, we asked if extensions of the gRNA at its 5' end would increase the collateral activity of enAsCas12a. Such extensions had been reported to increase the gene editing efficiency of Cas12a in cells and in vivo[63]. We extended the 5' end of the S2 gRNA by 4nt or 9nt (Fig. 7a). While the 4-nt extension did not improve the fluorescence signal in a *trans*-cleavage assay appreciably, the 9-nt extension did enhance the activity of enAsCas12a at 37 °C significantly (P < 0.001, one-sided Student's *t*-test) (Fig. 7b and Supplementary Fig. 31).

Third, we asked if gRNAs bearing extra chemical modifications would yield higher collateral activities with enAsCas12a than regular gRNAs. Specifically, we examined a guide targeting the S2 locus that contained 2'-O-methyl RNA bases, 2'-fluoro bases, and phosphorothioate linkages at various positions (Fig. 7a). The design was based on prior work that showed that such a guide yielded higher Cas12a editing efficiency than a regular gRNA in human cells[64]. We found that the extra chemical modifications did boost the rate of the Cas detection reaction at 37 °C significantly (P < 0.05, one-sided Student's *t*-test) (Fig. 7c and Supplementary Fig. 32).

Fourth, we assessed if DNA-RNA hybrid guides would give higher collateral activities with AsCas12a and its engineered variants than regular guides that contained only RNA bases. To this end, we generated S2-targeting guides that contained either two or four DNA base substitutions (Fig. 7a). We started with changes at the 3' end of the guide because that region of the guide complexed with its target appeared to be disordered in a previously solved crystal structure and thus might be flexible[65]. In addition, we tried substitutions at positions 1 and 8 in the spacer as those positions had previously been shown to tolerate mismatches[66]. At 37 °C, both our hybrid guides (with two or four DNA bases) were able to significantly increase the collateral activity of AsCas12a and its engineered variants relative to the original gRNA with no DNA bases (P < 0.05, one-sided Student's *t*-test) (Fig. 7d and Supplementary Fig. 33).

We sought to verify the effects of guide modifications using a different target site, the S6 locus. To this end, we generated three

new S6-targeting guides—one gRNA with a 4-nt 5' extension, one gRNA with a 9-nt 5' extension, and one hybrid guide with four DNA base substitutions (Fig. 7e). We decided not to pursue the chemically modified gRNA because it was much more expensive than a hybrid DNA-RNA guide but did not perform better. Overall, we found that the results from a *trans*-cleavage assay performed at 37 °C for the S6 locus mirrored those for the S2 locus (Fig. 7f and Supplementary Fig. 34). Extending the gRNA by 9nt at the 5' end or replacing four RNA bases with DNA bases significantly increased the collateral activity of enAsCas12a (P < 0.01, one-sided Student's *t*-test).

Since enAsCas12a appeared to be active over a wide range of temperatures, we next sought to evaluate the modified guides at 60 °C. For the S2-targeting set of guides, we observed that both gRNAs with 5' extensions as well as both hybrid DNA-RNA guides exhibited faster reaction kinetics with enAsCas12a than the original unmodified gRNA; by 5 minutes, they generated significantly higher fluorescence signals in a *trans*-cleavage assay (P < 0.001, one-sided Student's *t*-test) (Fig. 7g and Supplementary Fig. 35). Unexpectedly, however, the two gRNAs with 5' extensions triggered the collateral activity of enAsCas12a even in the absence of a template, as shown by the obvious increase in background signal by 30 minutes of reaction time (Fig. 7h and Supplementary Fig. 35). Similar results were obtained with three different reaction buffers. Moving forward, we dropped these two gRNAs from further consideration. For the S6-targeting set of guides, the results obtained at 60 °C mirrored those obtained at 37 °C (Fig. 7g, h and Supplementary Fig. 36). Both the gRNA with a 9-nt 5' extension and the hybrid guide increased the rate of reaction significantly (P < 0.05, one-sided Student's *t*-test) and there was no unexpected triggering of enAsCas12's collateral activity in the absence of a template.

Subsequently, we asked how simultaneous deployment of two modified guides together with enAsCas12a would improve the CRISPR detection module. We benchmarked the original set of unmodified S2 and S6 gRNAs against the S2 hybrid guide containing four DNA bases combined with either the S6 gRNA extended by 9nt at its 5' end or the S6 hybrid guide containing four DNA bases (Fig. 7i). In the presence of the intended SARS-CoV-2 template, each set of modified guides exhibited faster reaction kinetics than the original set of unmodified gRNAs, with the fluorescence signal saturating within 5 min. Furthermore, we observed that the modified guides completely suppressed any collateral activity of enAsCas12a in the absence of a template or in the presence of the closely related SARS-CoV and MERS-CoV templates. Collectively, our results demonstrate that the use of modified guides can increase the rate of the Cas detection reaction and effectively curb any off-target activity.

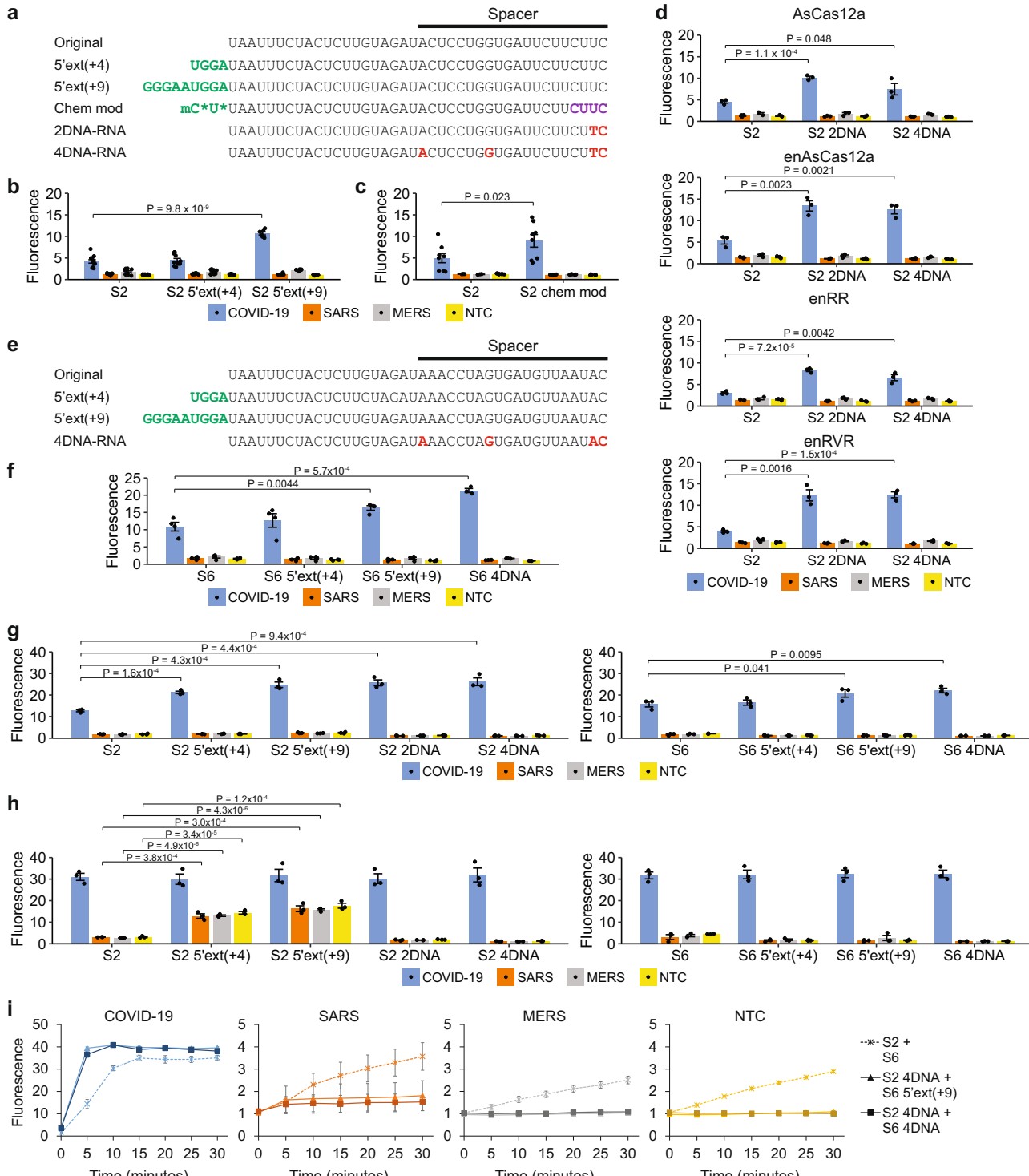

**Evaluation of a quasi-one-pot reaction with clinical RNA samples**. We wondered how we could improve the complete RT-LAMP-CRISPR workflow. So far, we had been transferring only 4 µl of LAMP products into 46 µl of CRISPR reaction mix to minimize a change in buffer of the Cas detection step. However, the LAMP reaction itself had a total volume of 25 µl, which was not being utilized fully. Hence, we tested if the CRISPR reaction was able to tolerate a larger amount of unpurified LAMP products, while keeping its final volume constant at 50 µl (Supplementary Fig. 37). We found that the kinetics of the CRISPR reaction gradually became slower with an increasing amount of

LAMP products in it. The fluorescence signal for a setup containing the entire 25 µl LAMP mix in the CRISPR reaction (i.e. the LAMP products were diluted 1:1) was significantly lower than that for the original workflow at both 37 °C and 60 °C ($P < 0.05$, one-sided Student's $t$-test). This suggested that some unknown factor in the LAMP mix might be partially inhibitory to the enAsCas12a enzyme and had to be diluted out. We then tested an alternative setup where instead of transferring LAMP products into the CRISPR reaction, we added 50 µl of CRISPR reaction mix into the LAMP reaction tube instead (i.e. the LAMP products were diluted 1:2). This not only enabled us to utilize all the LAMP

**Fig. 7 Guide engineering to enhance the CRISPR detection module. a** Sequences of the original S2-targeting gRNA and the modified guides evaluated in our work. 5′ extensions are indicated in green. 2′-O-methyl ribonucleotides (2′OMe RNA) are indicated by an extra lower-case m before the relevant nucleotide. 2′-deoxy-2′-fluoro-ribonucleotides (2′F RNA) are indicated in purple. DNA nucleotides are indicated in red. Phosphorothioate (PS) bonds are marked by asterisks. **b** Comparison of 5′-extended gRNAs with the original S2-targeting gRNA at 37 °C for enAsCas12a. Fluorescence measurements here were taken using a microplate reader after 5 min of cleavage reaction with 2E11 copies of synthetic DNA. Data represent mean ± s.e.m. ($n = 6$ [S2 5′ext (+9)] or 11 [S2, S2 5′ext(+4)] biological replicates). P-value was calculated using one-sided Student's t-test. **c** Comparison of a chemically modified gRNA with the original S2-targeting gRNA at 37 °C for enAsCas12a. Data represent mean ± s.e.m. ($n = 3$ [S2 SARS, S2 MERS], 6 [S2 chem mod SARS, S2 chem mod MERS], or 8 [COVID-19, NTC] biological replicates). P-value was calculated using one-sided Student's t-test. **d** Comparison of DNA-RNA hybrid guides with the original S2-targeting gRNA at 37 °C for various Cas12a enzymes. Data represent mean ± s.e.m. ($n = 3$ biological replicates). P-values were calculated using one-sided Student's t-test. **e** Sequences of the original S6-targeting gRNA and the modified guides evaluated in our work. 5′ extensions are indicated in green. DNA nucleotides are indicated in red. **f** Comparison of 5′-extended gRNAs and a DNA-RNA hybrid guide with the original S6-targeting gRNA at 37 °C for enAsCas12a. Data represent mean ± s.e.m. ($n = 3$ [S6 4DNA] or 4 [S6, S6 5′ext(+4), S6 5′ext(+9)] biological replicates). P-values were calculated using one-sided Student's t-test. **g, h** Comparison of 5′-extended gRNAs and DNA-RNA hybrid guides with the corresponding unmodified gRNAs at 60 °C for enAsCas12a. The S2-targeting and S6-targeting guides were tested separately. Fluorescence measurements here were taken after **g** 5 min and **h** 30 min of cleavage reaction with 2E11 copies of synthetic DNA. Data represent mean ± s.e.m. ($n = 3$ biological replicates). P-values were calculated using one-sided Student's t-test. **i** Evaluation of our two-gRNA CRISPR module using unmodified and modified guides. Fluorescence measurements were taken every 5 minutes using a microplate reader. The enAsCas12a-mediated cleavage reaction was performed at 60 °C with 2E11 copies of synthetic DNA. Data represent mean ± s.e.m. ($n = 3$ [S2 + S6 negative controls, S2 4DNA + S6 5′ext(+9)] or 5 [S2 + S6 COVID-19, S2 4DNA + S6 4DNA] biological replicates). Source data are available in the Source Data file.

products but also helped to minimize human error and cross-contamination, since nothing was taken out of the LAMP reaction tube. Encouragingly, we found that the alternative setup exhibited similar reaction kinetics to the original workflow and gave an even higher fluorescence signal at saturation.

Subsequently, we sought to compare the LoD of the original workflow and the alternative setup using synthetic SARS-CoV-2 RNA (Fig. 8a, b and Supplementary Fig. 38). RT-LAMP was performed at 65 °C, while the Cas detection reaction was performed at 60 °C. While only one out of three replicates showed successful amplification of two copies of viral template in the original workflow, all replicates were successful in the alternative setup. We repeated the experiments with different buffers for the CRISPR reaction and obtained similar results. Furthermore, we verified that a small amount of UTM could be tolerated even when all the LAMP products were utilized in the downstream Cas detection step (Supplementary Fig. 39). Hence, moving forward, we adopted the alternative setup, where 50 µl of enAsCas12a RNPs in Tango buffer was added directly into the LAMP reaction tube after completion of isothermal amplification.

While our project was ongoing, another study was published reporting that the speed and sensitivity of LAMP could be enhanced by guanidine[67]. Hence, we sought to determine whether guanidine or glycine, which we had incorporated into our CRISPR-Dx earlier (Fig. 4b), would be better for the assay. We first performed RT-LAMP alone and found that guanidine increased the reaction rate more appreciably than glycine (Fig. 8c). Next, we performed the entire assay with either guanidine or glycine in the reaction mix and observed that the assay with guanidine appeared to be more sensitive (Fig. 8d and Supplementary Fig. 40). Guanidine enabled 10 viral copies to be detected in seven out of eight replicates, while glycine enabled successful detection in only two out of eight replicates. Hence, we replaced glycine with guanidine in our assay and confirmed the speed and sensitivity of our updated test using dipsticks (Fig. 8e,f).

The surprising robustness of enAsCas12a to temperature afforded us an opportunity to perform the entire RT-LAMP-CRISPR workflow in a single temperature step. RT-LAMP had hitherto been performed at 65 °C, while the Cas detection reaction had only been tested up till 60 °C. Therefore, we wondered if both stages could be carried out at the same temperature. We first tested the CRISPR reaction at 60, 63, and 65 °C and observed that while the fluorescence readout only

decreased slightly at 63 °C, the drop in signal was more appreciable at 65 °C (Supplementary Fig. 41a). Next, we evaluated the sensitivity of our assay with RT-LAMP performed at a slightly lower temperature of 63 °C, while maintaining the CRISPR step at 60 °C. 14 out of 15 replicates showed successful amplification of 20 copies of synthetic viral template (Supplementary Fig. 41b). Finally, we tested if the whole workflow could be performed with just one heat block set at either 63 °C or 60 °C. The duration of RT-LAMP was extended from 15 to 22 min to accommodate for the somewhat sub-optimal temperature faced by the isothermal amplification reaction. Remarkably, our lateral flow assays revealed that positive test results could be obtained within 5 min of CRISPR reaction even with only two copies of synthetic viral template in the sample (Supplementary Fig. 41c, d). We termed our single heat block setup "quasi-one-pot", where the enAsCas12a RNPs were added directly into the LAMP reaction tube without the sample changing temperature. Notably, the entire assay can be completed within 30 min (22 min for RT-LAMP, 5 min for the *trans*-cleavage reaction, and 2 min for bands to develop on the dipsticks).

We asked if the quasi-one-pot setup would still be robust to SNVs at the target sites but yet exhibit exquisite specificity for SARS-CoV-2. To this end, we examined the mismatch tolerance of our optimized assay (enAsCas12a complexed with two DNA-RNA hybrid guides) and found that it showed similar sensitivity for the wild-type and the S254F N234N double mutant template (Fig. 8g and Supplementary Fig. 42). We further tested our assay against a set of coronaviruses and other respiratory viruses, including influenza viruses, paramyxoviruses, and enteroviruses. Fluorescence was detected only for SARS-CoV-2 over the course of 30 min, thereby confirming the specificity of our test (Fig. 8h).

Subsequently, we subjected our CRISPR-Dx to clinical evaluation with RNA samples isolated from patient NP swabs, which had previously been analyzed by qRT-PCR in the hospital. These samples came from 45 patients with COVID-19 and 30 uninfected individuals. Similar to our earlier pilot test (Fig. 6d, e), all samples that were negative by qRT-PCR also emerged negative in the lateral flow assay, confirming a 100% specificity for our assay (Fig. 8i). In addition, our VaNGuard test returned an unambiguous positive result for clinical samples that had a Ct value of 33.32 or lower in qRT-PCR analysis (Fig. 8i, j). Hence, based on these clinical RNA samples, our assay exhibited a LoD of 50 copies per reaction or 2 copies per microliter.

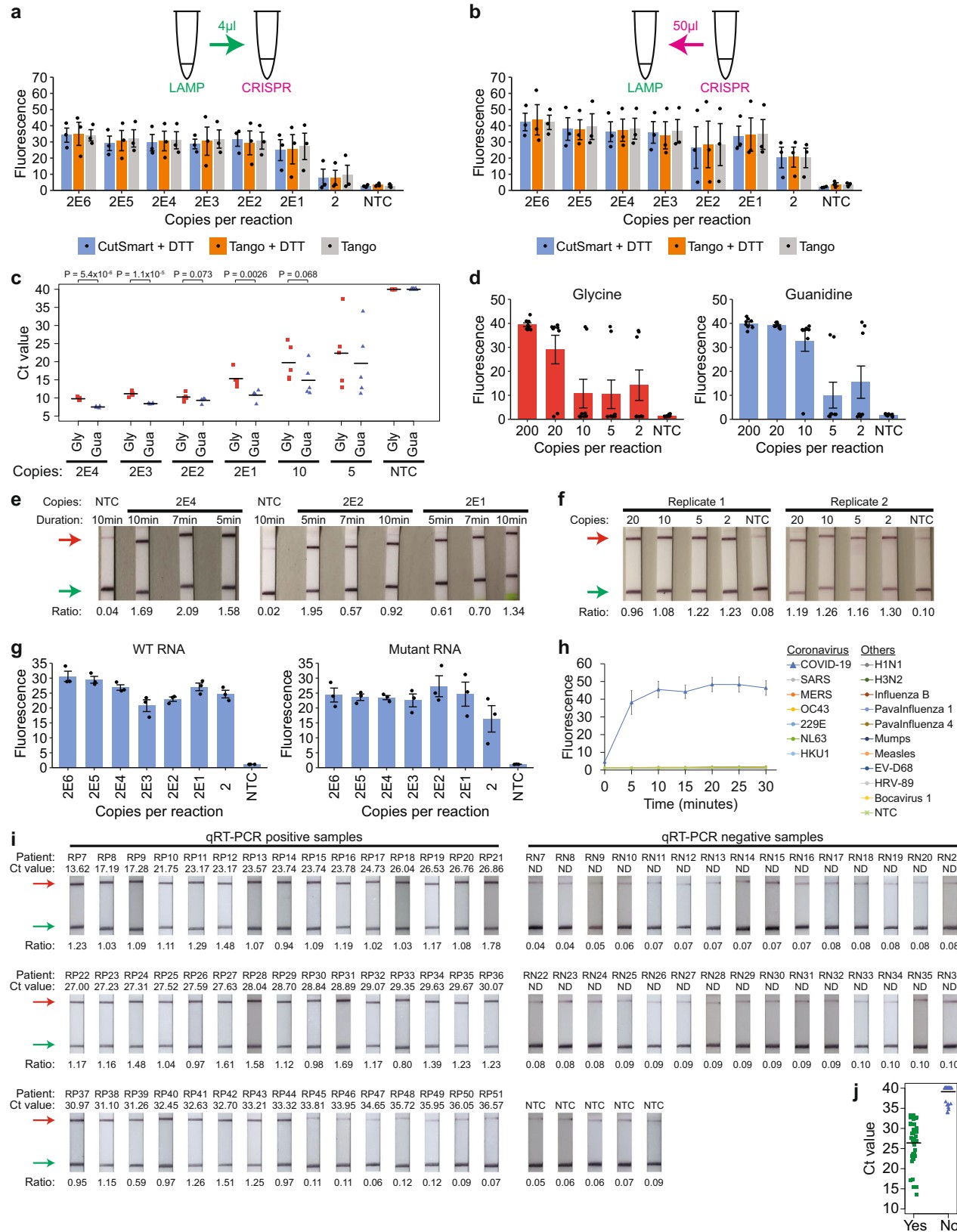

**Direct application of VaNGuard test on unpurified clinical samples.** An important consideration for rapid diagnostic tests is whether they can accept patient samples directly. RNA extraction usually takes at least 15 min and adds complexity to the workflow, thereby increasing the waiting time and making the test less usable by untrained professionals. Hence, we asked

if we could use our assay on patient samples directly without an additional RNA isolation step. One problem with patient samples is the presence of RNases that can rapidly degrade the viral RNA that we want to detect. To inactivate RNases, we first tried the Hudson protocol[16], but found that addition of TCEP and EDTA triggered spurious template-free amplification at Ct

**Fig. 8 Implementation of a quasi-one-pot reaction. a** Analytical LoD based on the original workflow. Different copies of synthetic SARS-CoV-2 RNA were used as input to RT-LAMP, which was performed at 65 °C for 15 min. The cleavage reaction was then carried out on only 4 μl LAMP products at 60 °C, with fluorescence measurements here taken after 10 min using a microplate reader. Data represent mean ± s.e.m. ($n = 3$ biological replicates). **b** Analytical LoD when all LAMP products were utilized. After RT-LAMP was completed, 50 μl CRISPR reagents were added directly into each sample tube for the cleavage reaction. Data represent mean ± s.e.m. ($n = 3$ biological replicates). **c** Strip chart showing how LAMP sensitivity was altered by substituting 0.1 M glycine (Gly) with 40 mM guanidine (Gua). RT-LAMP was performed at 65 °C in a real-time instrument with variable copies of synthetic RNA. The black horizontal bars among the data points in the strip chart represent the mean ($n = 5$ biological replicates). *P*-values were calculated using one-sided Student's *t*-test. **d** Comparing the assay sensitivity between glycine and guanidine. Different copies of synthetic SARS-CoV-2 RNA were used as input to RT-LAMP, which was performed at 65 °C for 15 min. Subsequently, 50 μl CRISPR reagents were added directly into each sample tube and the cleavage reaction was carried out at 60 °C, with fluorescence measurements here taken after 10 min using a microplate reader. Data represent mean ± s.e.m. ($n = 8$ biological replicates). **e** Lateral flow assays to assess cleavage reaction kinetics with guanidine in the assay mix. The enAsCas12a enzyme was complexed with both the S2 hybrid guide and the S6 5′-extended gRNA. After RT-LAMP was completed, the Cas detection reaction was performed at 60 °C for 5, 7, or 10 min before a dipstick was added to each sample tube. **f** Lateral flow assays to evaluate VaNGuard test sensitivity with guanidine in the assay mix. Different copies of synthetic SARS-CoV-2 RNA were used as input to RT-LAMP, which was performed at 65 °C for 15 min. The Cas detection reaction was then carried out at 60 °C for 5 min before a dipstick was added to each sample tube. **g** Analytical LoD for WT or S254F N234N double mutant RNA template using a quasi-one-pot reaction. The enAsCas12a enzyme was complexed with both the S2 and S6 hybrid guides. Fluorescence measurements here were taken after 5 min of *trans*-cleavage reaction. Data represent mean ± s.e.m. ($n = 3$ biological replicates). **h** Evaluating the specificity of our VaNGuard test. The enAsCas12a enzyme was complexed with both the S2 and S6 hybrid guides. 1E6 copies of synthetic RNA from different respiratory viruses were used as input to the quasi-one-pot reaction. Fluorescence measurements were taken at 5-min intervals using a microplate reader. Data represent mean ± s.e.m. ($n = 3$ biological replicates). **i** Evaluation with clinical RNA samples. Ct values were obtained using the Fortitude Kit. The enAsCas12a enzyme was complexed with both the S2 and S6 hybrid guides. 2 μl of each RNA sample was used as input to the quasi-one-pot reaction. The Cas detection reaction was performed for 5 min before a dipstick was added to each sample tube. A ratio of less than 0.15 was considered to be negative in our test. Hence, for the purified RNA samples, the lateral flow assay gave 0 false positives and 8 false negatives (RP6, RP45-51). **j** Strip chart summarizing the results from the clinical evaluation of our VaNGuard test using purified RNA samples. "Yes" indicates that the samples emerged positive in our test, while "No" indicates that the samples emerged negative. Source data are available in the Source Data file.

values less than 25 in most of the replicates (Supplementary Fig. 43).

We explored other options for RNase inactivation, namely proteinase K treatment and heat[68,69]. As simulation, we generated a lentivirus that expressed the relevant S-gene fragment from SARS-CoV-2 and spiked different amounts of it into human saliva. We then performed the RT-LAMP reaction on the contrived specimens, which were left untreated, heated at 95 °C for 5 min only, or treated with both proteinase K and heat (Fig. 9a). In agreement with recent studies[68,69], we found that treatment with both proteinase K and heat appeared to improve the speed and sensitivity of RT-LAMP. Next, we assessed the LoD of our CRISPR-Dx using the contrived specimens as input. The samples were pre-treated with proteinase K and heat. Notably, our assay was able to detect 40 copies of the lentivirus in all replicates (Fig. 9b). Importantly, omission of an RNA extraction step did not affect the speed of our test, with the fluorescence signal saturating after 5 min of CRISPR reaction.

Next, we asked if our assay could detect SARS-CoV-2 virions in sample collection medium. To this end, we spiked different amounts of the virus produced by Vero E6 cells into clinically negative UTM, which had previously been used to collect swabs from healthy individuals. After proteinase K and heat treatment, we applied our assay on these contrived specimens and observed clear test bands on the dipsticks for 100 or more copies of SARS-CoV-2 (Fig. 9c).

Subsequently, we sought to evaluate our assay using clinical NP swabs directly without any RNA extraction. We obtained 21 samples from patients with COVID-19 infection and another 21 samples from healthy controls. Part of these samples had previously undergone RNA extraction and qRT-PCR testing in a diagnostic laboratory, so we could compare our test results with the Ct values given by the laboratory. We treated the samples with proteinase K and heat before performing the lateral flow assay (Fig. 9d). Expectedly, all the clinically negative specimens also turned out to be negative in our VaNGuard test, verifying its 100% specificity. In addition, the test returned an unambiguous positive result for clinical samples that had a Ct value of 28.98 or

lower, which corresponded to 1000 or more copies per reaction. Curiously, we also observed that while our test appeared to have missed a sample with Ct value of 29.42, it correctly flagged another sample with Ct value of 30.36. Hence, to increase the likelihood of detecting the virus, we re-tested the five clinically positive samples that had been misclassified with double the reaction volume and twice the amount of sample input (Fig. 9e). However, our test correctly identified only one extra sample, which had a Ct value of 31.80. Overall, unlike the earlier clinical evaluation with purified RNA samples (Fig. 8i, j), the boundary for the Ct value between a positive and a negative outcome in our test was not as clear-cut for the crude NP swab samples (Fig. 9f). For Ct values between 29 and 32, our assay may return a positive or a negative result. This ambiguity may be due to the unknown and potentially complex sample matrix (for example, mucus from the nose), which can vary from specimen to specimen and exert some inhibitory effect on the assay enzymes. Moreover, the extent of viral RNA recovery during the purification process in the diagnostic laboratory may not be perfectly consistent especially for samples with low viral loads. Therefore, a specimen that actually had a higher viral load in the original NP swab might end up having a poorer Ct value due to greater sample loss. Taken together, our results indicate that although more challenging, our VaNGuard test can be applied directly on patient samples without additional RNA purification, with a LoD of 1000 copies per reaction or 40 copies per microliter. As the swabs used in our study were collected in 3 ml of UTM and we took only 2 μl for our test, the sensitivity may be better if the swabs had been collected in a smaller volume of medium.

**Incorporating a human internal control within the same reaction.** A diagnostic test for COVID-19 should include a human internal control to verify that a negative result is due to an absence of the virus and not simply due to insufficient sample input. To this end, we sought to identify a suitable set of LAMP primers targeting some housekeeping gene to use in our assay. We screened three primers sets against *POP7*, four primer sets against *ACTB*, and four primer sets against *GAPDH*, using

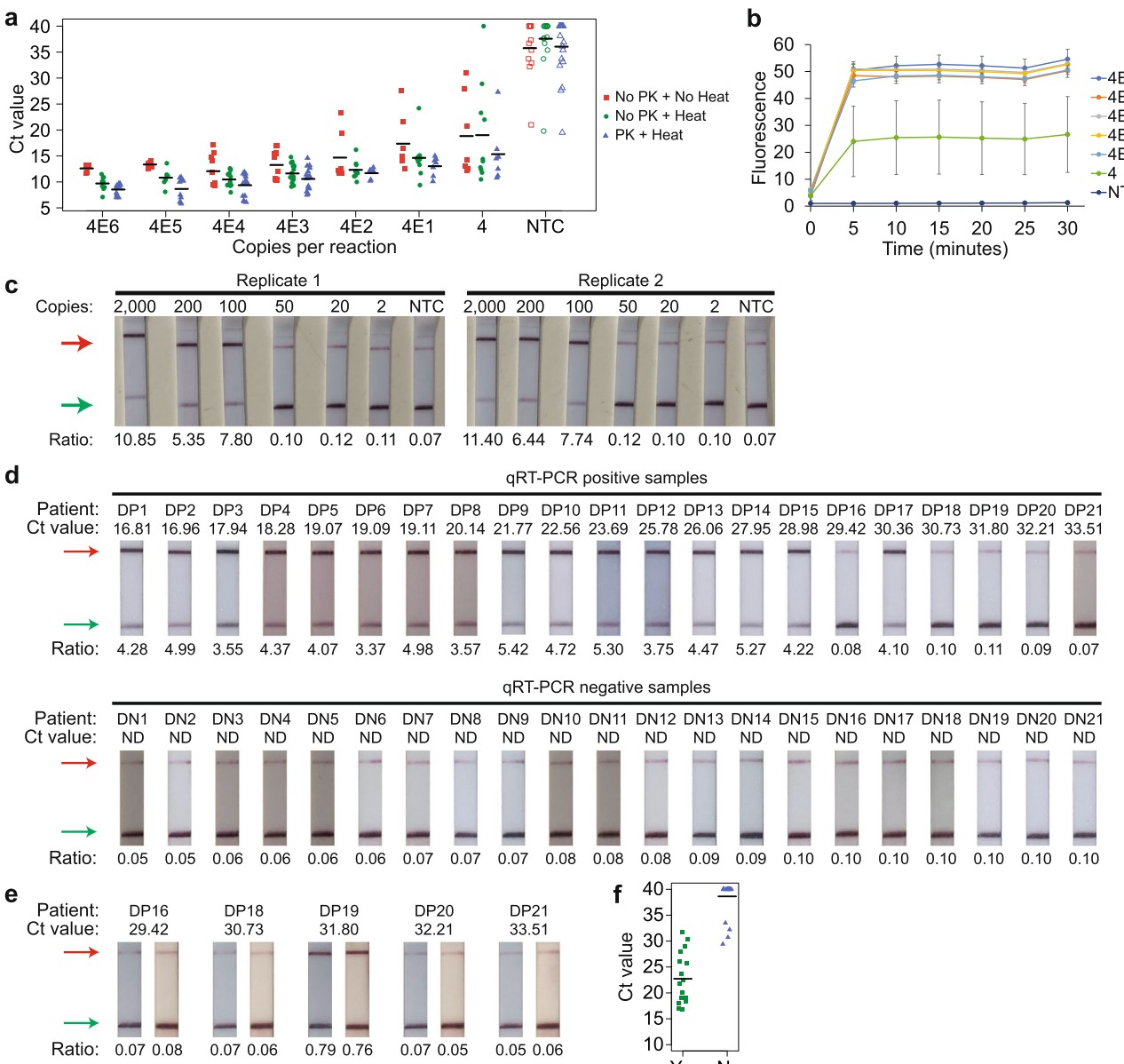

**Fig. 9 Application of our VaNGuard assay on crude samples. a** Strip chart showing the effect of proteinase K and heat treatment on RT-LAMP when different copies of S-gene-expressing lentivirus spiked into saliva were used as input. The black horizontal bars among the data points in the strip chart represent the mean (no PK + no heat: n = 5 [4E5-4E6], 7 [4-4E2], 10 [4E4], or 12 [4E3 and NTC]; no PK + heat: n = 7 [4E5], 9 [4E6], 10 [4-4E2], 15 [4E4], or 17 [4E3 and NTC]; PK + heat: n = 7 [4-4E2], 11 [4E5], 13 [4E6], 15 [4E4], 18 [4E3], or 20 [NTC] biological replicates). **b** Evaluating VaNGuard test sensitivity to unpurified pseudovirus. The enAsCas12a enzyme was complexed with both the S2 and S6 hybrid guides. Different copies of lentivirus spiked into saliva were treated with proteinase K and heat before being used as input to the quasi-one-pot reaction. Fluorescence measurements were taken at 5-minute intervals using a microplate reader. Data represent mean ± s.e.m. (n = 4 biological replicates). **c** Evaluating VaNGuard test sensitivity to unpurified SARS-CoV-2. The enAsCas12a enzyme was complexed with both the S2 and S6 hybrid guides. Different copies of the coronavirus produced in Vero E6 cells were spiked into clinically negative UTM, treated with proteinase K and heat, and then used as input to the quasi-one-pot reaction. The Cas detection step was carried out for 5 min before a dipstick was added to each reaction tube. **d** Clinical evaluation with NP swab samples. A Ct value of 30 was estimated to be equivalent to 500 copies of the virus. The enAsCas12a enzyme was complexed with both the S2 and S6 hybrid guides. Each sample was treated with proteinase K and heat before 2 µl was used as input to the quasi-one-pot reaction. The Cas detection step was carried out for 5 min before a dipstick was added to each reaction tube. **e** Re-test of misclassified NP swab samples using twice the reaction volume. 4 µl of each sample was used as input to the quasi-one-pot reaction. Overall, for the direct patient samples, the lateral flow assay gave 0 false positives and 4 false negatives (DP16, DP18, DP20, and DP21). **f** Strip chart summarizing the results from the clinical evaluation of our VaNGuard test using unpurified NP swab samples. "Yes" indicates that the samples emerged positive in our test, while "No" indicates that the samples emerged negative. Source data are available in the Source Data file.

heat-treated human saliva as sample input (Fig. 10a). All primer sets gave amplification products successfully, albeit at different rates. Furthermore, a few primer sets also yielded spurious by-products without a template.

Ideally, the internal control should be built into the same reaction tube as the COVID-19 test. Hence, we next evaluated if the human primers would interfere with our SARS-CoV-2 primers in the RT-LAMP reaction (Fig. 10b and Supplementary

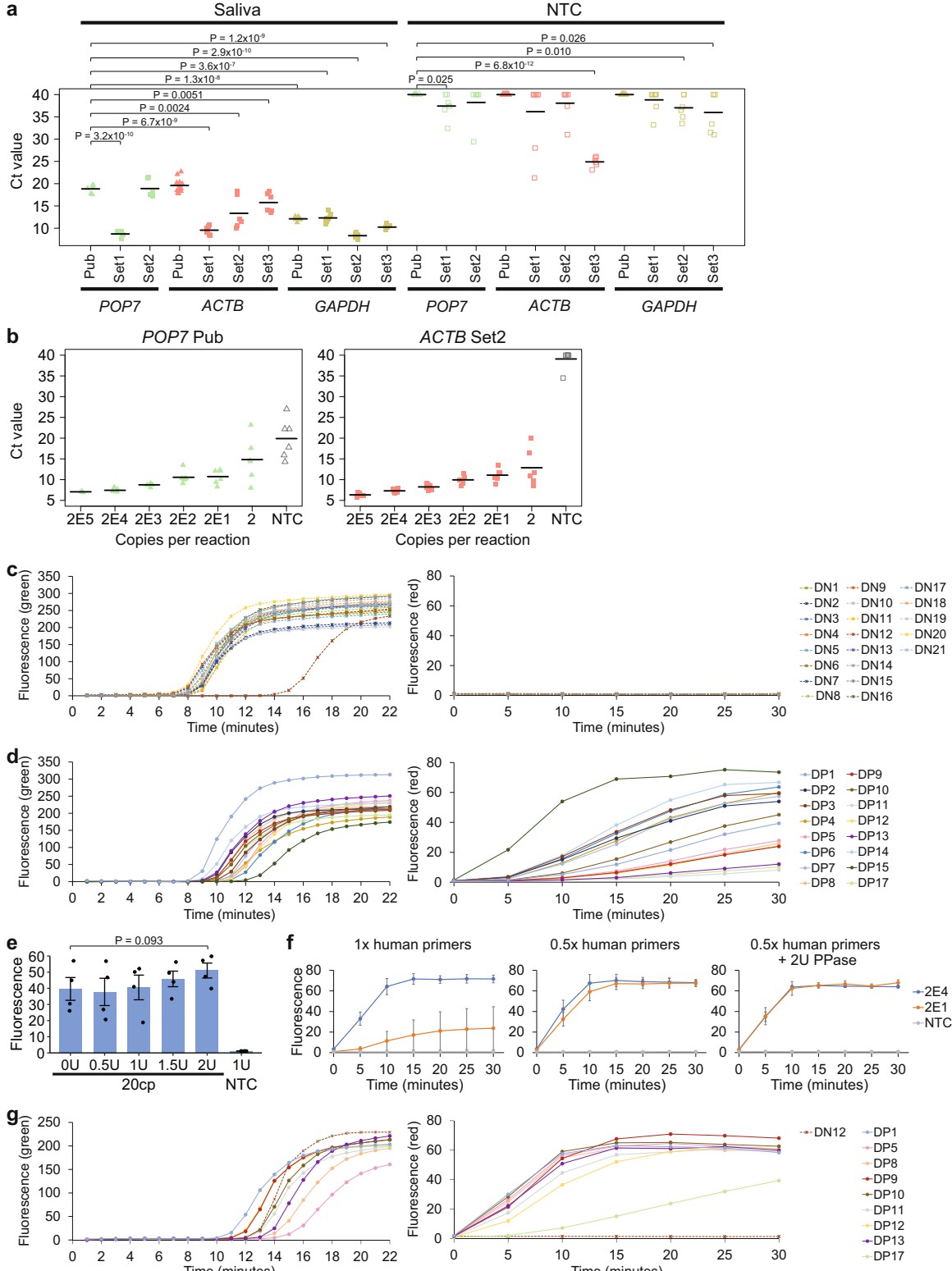

Fig. 44). Variable copies of synthetic viral RNA template were used. Interestingly, we observed that a few primer sets, such as the *POP7*-targeting primers deployed in the DETECTR system[17], caused non-specific amplification. We selected an *ACTB*-targeting primer set to proceed with because it did not trigger any serious mis-amplification without template and our SARS-CoV-2 LAMP primers continued to amplify well in its presence even at low copy numbers of viral RNA.

To incorporate an internal control within the same reaction tube as the COVID-19 test, a fluorescence readout has to be used instead of dipsticks and two colours are required to distinguish SARS-CoV-2 amplicons from human amplicons. We first replaced the green fluorophore (FAM) in the CRISPR reporter with a red fluorophore (Cy5) and verified that the signal was similar (Supplementary Fig. 45a). We then checked the sensitivity of our assay using synthetic viral RNA spiked into heat-treated

**Fig. 10 Development of a human internal control for our VaNGuard test. a** Strip chart showing the efficacy of different sets of LAMP primers targeting the human *POP7*, *ACTB*, or *GAPDH* gene. The primers labelled with "Set1", "Set2", or "Set3" are newly designed, while the primers labelled with "Pub" have been published[17,72,73]. 2 µl heat-treated saliva was used as sample input to RT-LAMP, which was performed at 65 °C over 40 min in a real-time instrument. The black horizontal bars among the data points in the strip chart represent the mean (*n* = 6 [*POP7* Pub and Set1-2, *ACTB* Set2-3, *GAPDH* Set2-3], 8 [*ACTB* Set1, *GAPDH* Pub and Set1], or 14 [*ACTB* Pub] biological replicates). Comparisons were done relative to the *POP7* Pub primers[17]. *P*-values were calculated using one-sided Student's *t*-test. **b** Strip chart showing the effect of different human primer sets on isothermal amplification of the SARS-CoV-2 S-gene. Different copies of synthetic SARS-CoV-2 RNA were used as sample input to RT-LAMP, which was performed at 65 °C over 40 min in a real-time instrument. The black horizontal bars among the data points in the strip chart represent the mean (*n* = 3 [*POP7* Pub 2E5] or 6 [*POP7* Pub all except 2E5, *ACTB* Set2] biological replicates). **c, d** Evaluation of our prototype VaNGuard assay containing a human internal control using **c** clinically negative and **d** clinically positive NP swab samples. The green fluorescence originates from a generic DNA-binding dye, while the red fluorescence originates from a Cy5-quencher reporter specific for SARS-CoV-2. Each sample was treated with proteinase K and heat before 2 µl was used as input to the quasi-one-pot reaction. **e** 20 copies of synthetic SARS-CoV-2 RNA were used as input to the quasi-one-pot reaction with different amounts of pyrophosphatase added during the Cas detection step. The fluorescence measurements here were taken after 5 min of *trans*-cleavage reaction. Data represent mean ± s.e.m. (*n* = 4 biological replicates). *P*-value was calculated using one-sided Student's *t*-test. **f** Evaluation of our assay with various amounts of human primers and pyrophosphatase. Different copies of synthetic RNA spiked into heat-treated saliva were used as input to the quasi-one-pot reaction. Fluorescence measurements were taken at 5-min intervals using a microplate reader. Data represent mean ± s.e.m. (*n* = 3 biological replicates). **g** Clinical evaluation of our optimized VaNGuard assay containing a human internal control. 2 µl of each proteinase K- and heat-treated NP swab sample was used as input. Source data are available in the Source Data file.

saliva. The reaction mix contained a generic green DNA-binding dye and the Cy5-reporter. Encouragingly, we observed that our assay could detect 20 or more copies of viral RNA even with concomitant amplification of human *ACTB* (Supplementary Fig. 45b). Furthermore, green, but not red, fluorescence was detected for the no-template control reaction as desired.

Subsequently, we sought to evaluate our assay with the internal control on unpurified clinical samples. To this end, we utilized the same set of NP swabs that was evaluated earlier using dipsticks (Fig. 9d–f). For all the qRT-PCR negative samples, amplification was observed for human *ACTB* but not for the S-gene of SARS-CoV-2 as expected (Fig. 10c). For the qRT-PCR-positive samples, we focused on those that had been classified correctly in the original assay. Amplification was observed in both the green and red channels for every sample (Fig. 10d). However, the CRISPR reaction kinetics for many samples was slower than expected.

We sought to improve our assay with the internal control, using synthetic viral RNA spiked into heat-treated saliva in the troubleshooting experiments. During the LAMP reaction, a large amount of pyrophosphate is produced, causing magnesium to precipitate out of solution. We wondered whether this would reduce the concentration of magnesium ions available for reaction over time or whether the pyrophosphate was inhibitory to the CRISPR reaction. Hence, we tested if addition of a thermostable pyrophosphatase would improve our VaNGuard test. While addition of up to 2U pyrophosphatase did not appear to improve RT-LAMP (Supplementary Fig. 46), it did appreciably enhance the kinetics of the Cas detection reaction (Fig. 10e and Supplementary Fig. 47). Next, we hypothesized that the human primers were competing with the SARS-CoV-2 primers for LAMP reagents. Hence, we investigated the effect of halving the amount of human *ACTB* primers in our assay. Overall, reduction in the concentration of the human primers led to a perceivably slower rate of green fluorescence generation presumably due to less efficient amplification of *ACTB* (Supplementary Fig. 48). Nevertheless, it did not prevent any of the replicates from amplifying successfully within the RT-LAMP duration of 22 minutes. In addition, we discovered that usage of less human primers clearly enhanced the sensitivity of our assay for SARS-CoV-2 (Fig. 10f). Further addition of 2U pyrophosphatase in the Cas detection reaction also enhanced the kinetics of the reaction.

Finally, we wondered how the improved version of our assay with internal control would perform on real patient samples. We re-evaluated the clinically positive NP swabs that yielded weaker-

than-expected red fluorescence signals in our earlier dual-colour assay (Fig. 10d). Additionally, we re-analysed DN12 as it previously showed later amplification of *ACTB* than the other clinically negative samples (Fig. 10c) and thus we were concerned that halving the amount of human primers might prevent the internal control from working in this specimen. The re-test revealed that adjustment of LAMP primer concentrations and addition of 2U pyrophosphatase enabled more robust detection of SARS-CoV-2 in every clinically positive NP swab (Fig. 10g). Furthermore, amplification was still observed for the human *ACTB* gene, but not for the S-gene of SARS-CoV-2, in DN12. Taken together, the results demonstrate that our VaNGuard test containing an internal control within the same reaction can be successfully applied on crude clinical samples without RNA extraction.

## Discussion

Rapid diagnostic tests for COVID-19 are essential for minimizing human-to-human transmission of SARS-CoV-2 so that our societies can safely re-open. Among the different types of rapid tests that are currently under active development, CRISPR-Dx has emerged as a major class of assays that can help meet the increasing demand worldwide for fast diagnosis especially in point-of-need settings. Due to the ease-of-use of the technology, multiple CRISPR-based assays have already been announced within a few months of the pandemic (Supplementary Data 1)[17–40].

Although promising, existing CRISPR-based tests suffer from several shortcomings. First, they have not taken into account viral evolution and RNA editing mediated by the ADAR and APOBEC enzymes, which can introduce unexpected variations at target sites and impact upon the performance of COVID-19 diagnostics. As a case in point, mutations in the SARS-CoV-2 genome have been observed at the primer binding sites of several qRT-PCR diagnostic tests[45]. Second, CRISPR-based assays usually take around 40 min to complete. Although fast compared to qRT-PCR tests, the assay duration may still be too long as the waiting time in point-of-need settings should ideally be as short as possible. Hence, strategies that can reduce the assay duration are desired. Third, the Cas detection reaction is typically combined with an isothermal amplification step to boost sensitivity. LAMP[8] is the method-of-choice in the current pandemic climate, as its reagents are readily available from several suppliers. However, the optimal temperature for most Cas enzymes deployed in diagnostic

applications is around 37 °C, while the LAMP reaction is usually carried out at 65 °C. Due to the temperature disparity, two heat blocks are required for many CRISPR-based tests, such as the DETECTR system[17]. Besides the increase in asset requirement, time is also wasted in cooling the samples between the two steps. To overcome the problem, some COVID-19 tests, such as AIOD-CRISPR[23], have relied on other isothermal amplification approaches like recombinase polymerase amplification (RPA)[7]. However, limitation in reagent supplies is a critical issue in a pandemic. We ordered an RPA kit at the start of the pandemic, but only received it 5 months later. Alternatively, some assay developers have tried to skip the amplification step[27], but this results in a big drop in test sensitivity. Hence, the ideal situation is to utilize a thermophilic Cas enzyme that can function well at 60 °C or above. However, we are not aware of such an enzyme besides the AapCas12b nuclease used in STOPCovid[22] and iSCAN[28] as well as the TtCsm complex used in CRISPR-Csm[39]. Fourth, for a rapid diagnostic test to be truly useful in a point-of-need setting, it should be able to accept patient samples directly. Although purified RNA is ideal for performance, the process of RNA extraction will take up precious time, increase cost, and stress the supply chain. Therefore, there is great interest in developing assays that can handle patient samples directly, including nasopharyngeal swabs and saliva. Fifth, most CRISPR-based assays have not incorporated a human internal control into the same reaction tube as the COVID-19 test. The control is required for real-life deployment to ensure that a negative test result is not simply due to an insufficient amount of sample input.

In this work, we developed a CRISPR-based assay that addressed the above problems. To bolster the robustness of our test against unexpected variant nucleotides introduced by evolutionary pressures or RNA editing, we implemented several distinct strategies. First, we tested several Cas12a enzymes and found that enAsCas12a exhibited the highest tolerance for SNVs at the gRNA-target interface. Second, we demonstrated that the use of two gRNAs (S2 and S6) with enAsCas12a further enhanced the robustness of our assay. Third, we incorporated truncated primers and a high-fidelity polymerase into the RT-LAMP reaction. With all these strategies in place, we showed that our VaNGuard test was able to detect low copies of viral RNA that harboured known mutations in some SARS-CoV-2 isolates from around the world. Nevertheless, further work is needed to determine how applicable the reported strategies are to other combinations of viral mutations and RNA editing events.

Besides robustness to SNVs in the viral genome, our VanGuard test also possesses other strengths. We found that the use of modified gRNAs, in particular hybrid DNA-RNA guides, accelerated the Cas detection reaction and suppressed any residual background activity to negligible levels. In addition, we discovered that enAsCas12a exhibited surprising robustness to reaction temperature and was active from 37 °C to over 60 °C. This enabled us to perform RT-LAMP and the Cas detection reaction in a single heat block at the same temperature. Furthermore, to maximize the real-world utility of our test, we demonstrated that it could be applied directly on crude clinical samples without any RNA extraction and we also successfully incorporated an internal control into the same reaction tube through the optimization of LAMP primer concentrations and the use of pyrophosphatase to reduce the built-up of pyrophosphate. The optimized VaNGuard test exhibits high specificity for SARS-CoV-2 and does not show any cross-reactivity with 16 other coronaviruses or respiratory viruses. To facilitate the interpretation of test results by a layperson, we developed a mobile phone app to analyse dipsticks (Supplementary Fig. 49) as well as designed and built a cheap do-it-

yourself device from discarded cartons that allowed fluorescence signals from the *trans*-cleavage assay to be readily visualized using light from a mobile phone (Supplementary Fig. 50). Overall, the cost of running the VaNGuard test is under S$10 per sample (Supplementary Data 3), which is around US$7.30 or €6.20. Bulk of the cost comes from the LAMP mastermix and the dipstick.

While our project was ongoing, a paper was published reporting that gRNAs with 3' extensions, but not 5' extensions, yielded more collateral cleavage in vitro than their unmodified counterparts[32]. However, we obtained opposite results in our work. Specifically, we found that gRNAs with UA-rich 3' extensions did not consistently enhance the *trans*-cleavage activity of Cas12a and in fact often gave poorer assay performance (Supplementary Fig. 30). In contrast, our S2 and S6 gRNAs with 9-nt 5' extensions produced appreciably more collateral cleavage instead (Fig. 7b, f, g), although we did observe that the 5' extended S2 gRNA triggered spurious template-free amplification at 60 °C for some unknown reason (Fig. 7h). Considering all the results, we speculate that gRNA extensions, regardless of whether they are at the 3' end or the 5' end, might disrupt RNA structure in unexpected ways and thus may not represent a generalizable strategy to enhance the efficiency of the CRISPR-Cas detection module in diagnostic applications. Consequently, in our final VaNGuard test, we have adopted guides whose overall sequences remained unchanged compared to wild-type but instead contained DNA nucleotide substitutions at specific locations within the spacer.

We note that diagnostic assays can be constructed out of isothermal amplification methods alone without coupling them to a separate CRISPR-Cas detection module. Such assays typically rely on the use of a turbidimeter to measure the extent of magnesium precipitation, labelled primers, or special dyes that sense pH changes, react with amplification by-products, or bind to double-stranded DNA. Due to their relative simplicity, numerous RT-LAMP-only diagnostic assays for COVID-19 have been developed and even commercialized. However, isothermal amplification frequently produces non-specific products without a template, giving rise to false positive results (Supplementary Fig. 51). Hence, the Cas detection step provides a valuable specificity check that rules out these undesirable false positives. Although one may also utilize an alternative sequence-specific detection probe that is distinct from the LAMP primers as a specificity check[70], the probe itself is not involved in any amplification process. In contrast, the CRISPR-Cas detection system is capable of signal amplification because each hyperactivated Cas nuclease can proceed to cleave numerous reporter molecules.

In conclusion, rapid diagnostic tests are essential for us to minimize viral transmission and re-open our societies safely. The availability of different types of rapid tests diversifies supply chains, thereby helping us to mitigate the risk of test shortages. CRISPR-Dx has emerged as one major type of rapid test. Our work here provides strategies for enhancing the robustness and speed of CRISPR-based assays and can also be adopted to fight disease X in the future.

## Methods

**Plasmids and oligonucleotides**. The pET28b-T7-Cas12a-NLS-6xHis expression plasmids were gifts from Keith Joung and Benjamin Kleinstiver (Addgene plasmid #114069 [AsCas12a], #114070 [LbCas12a], #114072 [enAsCas12a], #114075 [enRVR], and #114077 [enRR])[48]. DNA oligonucleotides, custom reporters for the *trans*-cleavage assays, and gene fragments (ORF1AB, S, and N) for the three coronaviruses SARS-CoV-2, SARS-CoV, and MERS-CoV were synthesised by Integrated DNA Technologies. PCR fragments of the S-gene and T2A-eGFP were cloned into a lentiviral vector using the NEBuilder HiFi DNA Assembly Kit (NEB). All oligonucleotides used in this study, including primers, are listed in Supplementary Data 4.

**Cas12a expression and purification.** The Cas12a expression plasmids were transformed into *Escherichia coli* BL21 (DE3) and stored as glycerol stocks. Starter cultures were grown in LB broth with 50 μg/ml kanamycin at 37 °C for 16 h and diluted 1:50 into 400 ml LB-kanamycin broth until an $OD_{600}$ of 0.4–0.6 was reached. Cultures were then induced with 1 mM isopropyl β-D-1-thiogalactopyranoside (IPTG) and incubated at 25 °C for another 16 h. Subsequently, cells were harvested by centrifugation at $3220 \times g$ for 20 min and resuspended in lysis buffer [50 mM HEPES, 500 mM NaCl, 2 mM $MgCl_2$, 20 mM imidazole, 1% Triton X-100, 1 mM DTT, 0.005 mg/ml lysozyme (Vivantis), 1X Halt Protease Inhibitor Cocktail (Thermo Fisher Scientific)], followed by sonication at high power for 10 cycles of 30 s ON/OFF (Bioruptor Plus; Diagenode). Lysates were clarified by centrifugation at $10,000 \times g$ for 15 min. The supernatants were pooled, loaded onto a gravity flow column packed with Ni-NTA agarose (Qiagen), and rotated for 2 h at 4 °C. The column was washed twice with 5 ml wash buffer (50 mM Tris, 300 mM NaCl and 30 mM imidazole). Five elutions were performed with 500 μl elution buffer (50 mM Tris, 300 mM NaCl, and 200 mM imidazole) and analysed by SDS-PAGE. The final gel filtration step was performed with a HiLoad 16/600 Superdex 200 pg column (GE Healthcare) on a fast protein liquid chromatography purification system (ÄKTA Explorer; GE Healthcare), which was eluted with storage buffer (50 mM Tris, 300 mM NaCl, and 1 mM DTT). Fractions containing Cas12a were collected, analysed by SDS-PAGE, and concentrated to around 500 μl with Vivaspin 20, 50,000 MWCO concentrator units (Sartorius). Glycerol was added to a final concentration of 20%. Protein concentrations were measured with the Quick Start Bradford Protein Assay (Bio-Rad) before the purified proteins were aliquoted and stored at −80 °C.

**gRNA design.** Complete genomes of SARS-CoV-2 (accession MN908947.3), SARS-CoV (accession NC_004718.3), MERS-CoV (accession NC_019843.3), CoV OC43 (accession NC_006213.1), CoV 229E (accession NC_002645.1), CoV NL63 (accession JX504050.1), and CoV HKU1 (accession KF686346.1) were retrieved from NCBI (https://www.ncbi.nlm.nih.gov/) and aligned with MUSCLE (https://www.ebi.ac.uk/Tools/msa/muscle/) using default settings. Potential target sites (20nt spacers) in the ORF1AB, S, and N genes were selected from non-conserved regions containing a TTTV PAM. Potential targets were filtered after a specificity check on BLASTn (https://blast.ncbi.nlm.nih.gov/Blast.cgi) to remove non-specific candidates. Truncated gRNAs were generated by shortening their spacers to 18nt and 19nt lengths at the 3' end.

**In vitro transcription (IVT) of gRNAs.** Templates for gRNA synthesis were designed with the following sequence order: T7 promoter-Cas12a scaffold-spacer. Top strand DNA oligos consisting of the T7 promoter (5'-TAATACGACTCAC TATAGG-3') and scaffold (5'-TAATTTCTACTCTTGTAGAT-3' for AsCas12a and its variants; 5'-AATTTCTACTAAGTGTAGAT-3' for LbCas12a) were annealed to the bottom strand and extended by Q5 High-Fidelity DNA polymerase (NEB). IVT of the dsDNA products was performed with the HiScribe T7 Quick High Yield RNA Synthesis kit (NEB) at 37 °C overnight. Following DNase I digestion, gRNAs were purified with the RNA Clean & Concentrator-5 kit (ZYMO Research), analysed by 2% TAE-agarose gel electrophoresis to assess RNA integrity, measured with NanoDrop 2000, and stored at −20 °C.

**Synthesis of DNA and RNA templates.** Gene fragments (gBlocks) were cloned into pCR-Blunt II-TOPO vector using the Zero Blunt TOPO PCR Cloning kit (Invitrogen) and their sequences were verified by Sanger sequencing. The vectors were used as templates for PCR with Q5 High-Fidelity DNA polymerase (NEB) and the products were gel extracted and purified with the PureNA Biospin Gel Extraction kit (Research Instruments). DNA concentrations were measured using NanoDrop 2000 and all the DNA samples were stored at 4 °C. To generate RNA templates for RT-LAMP assays, the forward primers used for PCR were appended with the T7 promoter sequence. After PCR amplification with the gBlock-TOPO vectors as template, IVT was performed as described for gRNA generation.

**RT-LAMP reaction.** Synthetic SARS-CoV-2 RNA templates were serially diluted and amplified using the WarmStart LAMP Kit (NEB). 10× S-gene LAMP primer mix was prepared with concentration of 2 μM for F3, 4 μM for B3, 8 μM for FIP (PM), BIP(PM), FIP(tPM-3), BIP(tPM-3), LF, and LB, and 16 μM for swarm F1c and swarm B1c. The RT-LAMP reaction containing 12.5 μl WarmStart LAMP Mastermix, 2.5 μl 10× S-gene primer mix, 2.5 μl 0.4 M guanidine HCl, 2.5 μl Q5 High-Fidelity Polymerase (0.06U/μL), and 5 μl synthetic RNA was then setup for a total reaction volume of 25 μl. Subsequently, the reaction tube was incubated at 65 °C for 22 min. For the assay with the internal control, RT-LAMP reactions also contained 0.5 μl LAMP dye (NEB) and primers targeting human *ACTB* with final concentration of 0.1 μM for F3 and B3, 0.8 μM for FIP and BIP, and 0.4 μM for LF and LB.

**Quasi-one-pot *trans*-cleavage assay.** For the lateral flow readout, the following components were combined together: 9 μl 541 nM Cas12a RNP, 7.5 μl 10× Tango buffer, 13.5 μl 500 nM FITC-biotin reporter, and 20 μl water. This 50 μl Cas12a reaction mix was then added directly into the 25 μl RT-LAMP reaction tube. Next, the reaction was incubated at 60 °C for at least 5 min. Subsequently, 75 μl

HybriDetect assay buffer (Milenia Biotec) was added to the reaction and a HybriDetect (Milenia Biotec) dipstick was inserted directly into the solution in an upright position. The dipstick was incubated in the reaction tube for 2 min at room temperature before inspection.

For the fluorescence readout, the following components were combined together, 9 μl 541 nM Cas12a RNP, 7.5 μl 10× Tango buffer, 1.5 μl 10 μM Cy5/ FAM-Quencher reporter, and 32 μl water. This 50 μl Cas12a reaction mix was then added directly into the 25 μl RT-LAMP reaction tube. Next, the reaction was incubated at 60 °C for 30 min and the fluorescence intensity was measured every 5 min using the Infinite M1000 Pro (Tecan), the Spectramax M5 plate reader (Molecular Devices), or the EnSpire Multimode Plate Reader (PerkinElmer).

**Evaluations with SARS-CoV-2 samples.** Heat-inactivated SARS-CoV-2 (ATCC VR-1986HK) was diluted into clinically negative Universal Transport Medium (UTM) (Copan) based on the droplet digital PCR (ddPCR) quantification provided by the vendor. Ethics approval for the use of leftover RNA patient samples was given by the National Healthcare Group Domain Specific Review Board (NHG-DSRB) (Study Reference Number: 2020/00867). For the NP swab samples, the research was waived for review by the A*STAR Institutional Review Board as the overall intent of the work was to develop a diagnostic assay to contribute to ongoing surveillance efforts, and control and preventive measures for the COVID-19 pandemic in Singapore. 8.3 μl of each NP swab sample was treated with 1 μl Proteinase K (NEB) and vortexed for 1 min at room temperature. The treated sample was then heated at 95 °C for 5 min before 2 μl was used for RT-LAMP.

**Reporting summary.** Further information on research design is available in the Nature Research Reporting Summary linked to this article.

## Data availability

Previously published whole genomes of SARS-CoV-2, SARS-CoV, MERS-CoV, CoV OC43, CoV 229E, CoV NL63, and CoV HKU1 are available in GenBank under the accession numbers MN908947.3, NC_004718.3, NC_019843.3, NC_006213.1, NC_002645.1, JX504050.1, and KF686346.1, respectively. Other datasets generated during the current study are available from the corresponding author on reasonable request. Source data are provided with this paper.

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

## Acknowledgements

This work is supported by an Industry Alignment Fund—Pre-positioning Programme grant from the Agency for Science, Technology and Research (A*STAR) (H18/01/a0/019

to M.H.T.), a Ministry of Education Academic Research Fund Tier 1 grant (2017-T1-001-214 to M.H.T.), an Open Fund—Individual Research Grant from the National Medical Research Council (NMRC/OFIRG/0017/2016 to M.H.T.), a Gap Fund from A*STAR (ACCL/19-GAP064-R20H-N to M.H.T.), a Gap Fund from Ministry of Education (NGF-2020-08-015 to M.H.T.), and core funding from the Genome Institute of Singapore (to M.H.T.). We also thank Xin Ning Nicole Tang for help with setting up the CRISPR diagnostics workflow, Sidney Yee and Maple Ye for reagents and advice, Xavier Roca for human cell line RNAs, Wei Leong Chew for the Twist Panel respiratory viral RNAs, and members of the DaRE lab for discussions.

## Author contributions

M.H.T. conceived the project and provided overall supervision. K.H.O., M.M.L., J.W.D.T., S.Y.T., P.K., and M.H.T. designed the experiments. K.H.O., M.M.L., J.W.D.T., S.Y.T., and P.K. performed the experiments. S.J. and Y.-G.G. assisted with protein purification. C.K.L., B.Y., and G.Y. assisted with clinical validations. S.M.-S. assisted with bioinformatics analysis of SARS-CoV-2 genomes in GISAID. J.H. and W.L. developed the mobile phone app. K.H.O., M.M.L., J.W.D.T., S.Y.T., P.K., and M.H.T. analysed the data. M.H.T. wrote the manuscript with inputs from K.H.O., M.M.L., J.W.D.T., S.Y.T., and P.K. All authors approved the manuscript.

## Competing interests

The authors declare that a patent has been filed for the VaNGuard test (Singapore Patent Application No. 10202006167 W). M.H.T., K.H.O., J.W.D.T., and S.Y.T. are co-inventors on the patent. The remaining authors declare no competing interests.
