## [Peer Review File · Nature Communications]

Reviewers' Comments:

Reviewer #1:

Remarks to the Author:

In this study, Ooi et al. undertook to produce a version on the CRISPR Cas12a-based SARS-CoV-2 detection assay that would be impervious to the variations in the viral RNA sequence that are bound to occur over time. In the process, they also tested and adjusted a number of parameters of the LAMP amplification step and the detection step itself. By combining a more mismatch-tolerant version of AsCas12a with two guide RNAs targeting a pair of sequences in the viral S gene, they largely accomplished their goal, while matching existing assays in sensitivity and specificity. In doing so, they maintained a clear focus on the needs of clinics and testing centers regarding simplicity, speed and cost.

Although my expertise does not lie in the area of clinical testing, this seems to me like a valuable contribution. The extensive testing of multiple variables was all done in a single lab, as far as I can see, so the robustness of the conclusions will still have to be tested more widely. Even if the parameters defined here do not precisely replicate elsewhere, important variables and approaches have been explored.

The presentation is quite clear, but I would suggest a few modifications.

- Move Supplementary Figure 15 to the main text and include the locations of the swarm primers. This will help readers follow the text in several places, particularly where the various amplification primers are being altered.
- It seems logical to put the sections on multiplexing LbCas12a and enAsCas12a together, rather than separating them with the sections on LAMP variables. The latter sections should probably precede the LbCas12a section.
- Provide statistics (Avg, StdDev) on the dipstick assays, particularly in cases where the signal above background is small – e.g., Figures 6b, 9f, 11e and S19c. This would increase confidence in those lower signals that are not obvious by eye.
- In the results section, the authors mention the advantage of relying on the commercially available LbCas12a, and they do so far as to demonstrate its use on mismatched viral templates. Then after an extensive section on optimizing enAsCas12a, they never return in the Discussion to how one might use the Lb enzyme when the enAs enzyme is not available. A concluding comparison of the two would be useful.

Reviewer #2:

Remarks to the Author:

In this manuscript, Ooi et al. tested the specificity of 5 different CRISPR/Cas12a variants (2 orthologs and 3 engineered AsCas12a) using the activators with mismatches. They then selected and combined two different orthologs of CRISPR/Cas12a to intentionally create a less-specific test for COVID-19 that tolerates mismatches and can detect the mutated spike protein genes of SARS-CoV-2 RNA. The authors combined an isothermal RT-LAMP step with CRISPR and reported a sensitivity of 20 copies/uL in a fluorescence-based assay for detecting the S gene. A lateral flow assay was also tested and validated with a handful of patient samples.

Novelty:

While similar fluorescence-based and paper-based tests using RT-LAMP+CRISPR/Cas12a tests have been developed and specificity of various CRISPR/Cas12a systems have been individually reported, the authors performed parallel comparisons of these variants for SARS-CoV-2 genes. They combined multiple low specificity variants of Cas12a to be able to detect both the wild-type and the mutated version of the test with similar efficiency. The authors performed optimization and analysis with different regions of SARS-CoV-2 genome to develop a COVID-19 test.

Major concerns:

1. I commend the authors on having doctors perform the blind test (see reporting summary). The authors tested their assay in a 6 positive and 6 negative patient samples and therefore it is not recommended to conclude 'clinically-validated' (see abstract). For instance, the FDA recommends testing at least 30 positive/30 negative samples as a guideline. Testing additional clinical samples will improve the robustness of the assay. I highly recommend testing additional samples.
2. "Nevertheless, while further work is needed to improve its clinical sensitivity, our assay is already an improvement over the DETECTR and Sherlock platforms in terms of analytical sensitivity and robustness against variant nucleotides." In the table 1 (or in a separate table), the authors should include the known test performance data of currently available methods such as qPCR, SHERLOCK, and DETECTR? The reported LOD is 20 copies per reaction. Please include LOD in copies/uL of patient sample for direct comparison. How does it truly compares with the DETECTR (Broughton et al., Nature Biotech, 2020). If the sensitivity is lower than reported, authors cannot claim an improved sensitivity. Furthermore, as authors commented in Lines 566-569 "Hence, our results indicate that despite its high analytical sensitivity of 20 copies per reaction, our VaNGuard assay appears to exhibit a lower clinical sensitivity possibly because the matrix of clinical samples may exert some inhibitory effect on the assay enzymes." Missing information about the patient samples. What sample type (nasal swabs, saliva) was chosen for this study? Since the true sensitivity might be even lower in the matrix and the true LOD should be calculated in the same matrix by pooling negative samples.
3. Since the virus mutations and viral RNA editing in the host are indeterministic in nature and therefore it is not possible to claim that the CRISPR/Cas12a combinations would still be functional with the same efficiency after the virus mutates. Please clarify.
4. In general, decreasing the specificity increases the sensitivity. However, combining different variants of Cas12a with different sensitivities can further lower the sensitivity of detection as shown in Figure 5. How do authors plan to maintain good sensitivity?
5. The authors included GISAID dataset for inclusivity analysis (Supplementary Dataset 1). However, to confirm the true applicability of VaNGuard for detecting mutated samples, the inclusivity testing should be performed with at least a couple of different isolates.
6. RT-LAMP is prone to false-positives and cross-contamination as shown by the authors (SI Fig. 27). Please indicate how the authors tried to resolved this issue. It is unclear from the figures.

Minor concerns:

7. There are some key CRISPR-based COVID-19 tests missing from table 1 and I recommend adding them for completeness. These include: SHERLOCK (Cas13a-based, the first one to get an FDA EUA), MB-Qdot CRISPR (Cas12a-based), CRISPR-ENHANCE (Cas12a-based), SHINE (similar to CARMEN, Cas13a-based), CRISPR-CADD (dCas9-based), IMPACT chip (Cas12a-based).
8. Is the enAsCas12a same as the E174R/S542R/K548R version. Please indicate that in the manuscript?
9. In the reporting summary, the statistics section with statistical test information is missing. In fact, most figures are missing the statistical analysis.
10. Line189 "There was no cross-reactivity for SARS-CoV or MERS-CoV."
Lines 204-205 "Importantly, no cross-reactivity for SARS-CoV or MERS-CoV was observed regardless of the Cas12a nuclease used."
Lines 225-226 "There was again no cross-reactivity for SARS-CoV and MERS-CoV. "
There are too much verbatim and no citing of the figures. The authors actually performed the bioinformatic analyses (Fig. S5) and experimental confirmation (Fig. 9a) for SARS-CoV and MERS-CoV. Please remove duplicate sentences and add proper call-outs to the figures.
11. Also, include the RT-LAMP primers in the alignment analysis and add it to the SI section. I also recommend aligning the sequences with all the other human coronaviruses HCoV-229E, HCoV-OC43, HCoV-NL63, and HCoV-HKU1 bioinformatically.
12. Line: 260-262 "Strikingly, we now observed that LbCas12a exhibited the best overall tolerance for mismatches, while AsCas12a and its engineered variants became more sensitive to the mismatches."
Why was AsCas12a chosen if the LbCas12a was the best? Please add to the discussion.

13. Line 159-160: "Further increase in temperature from 65oC to 68oC caused a deterioration in the performance of our CRISPR-Dx (Supplementary Fig. 4)".

Did the authors mean RT-LAMP instead of CRISPR-Dx?

14. Figure 2: Please clearly indicate on the plot that the 62oC, 65oC represents RT-LAMP temperatures.

15. For Cas12a, I would call 'crRNA' instead of 'gRNA'. Interestingly, the term 'crRNA' is not even mentioned in the entire manuscript.

Reviewer #3:

Remarks to the Author:

This paper by Ooi, Liu, Tay, Teo, and Kaewsapsak et al. describes a CRISPR-based diagnostic assay for COVID-19 called VaNGuard. The assay uses an engineered Cas12, which is more tolerant to mismatches than the wildtype Cas12. The authors demonstrate that their assay is sensitive and specific for SARS-CoV-2, and test a small number of patient samples.

Although some mutations have occurred in the SARS-CoV-2 genome, there are only a handful of sites where the minor allele frequency is high (as of September 18, approximately 10 sites have a minor allele frequency > 10%). Thus, the problem that the authors are setting out to address is relatively minor, consisting of improved performance on hypothetical viral sequences that may at some point emerge in the future.

This assay, as described, is very similar to the previously published DETECTR assay for SARS-CoV-2. The authors have made some minor changes to the assay (modifying individual nucleotides in the primer sequences, changing the RT-LAMP temperature, and using different Cas12 ortholog & guide RNAs).

Major comments:

1. The DETECTR assay published by Broughton et al. already uses multiple crRNAs targeting SARS-CoV-2 (N and E genes), so the concept of using multiple crRNAs is not novel. There is a broader lack of novelty in this paper, as most of the results shown in the figures are incremental or minor modifications of existing protocols.

2. The assay has reduced performance on viral seedstocks (Fig. 11d) relative to synthetic targets, by at least 1 order of magnitude (10-fold). There is a further reduction of sensitivity in Fig. 11e, where the authors are only able to detect patient samples down to ~190 copies. The authors speculate that this could be due to clinical matrix, but the samples consist of extracted RNA so this is not possible. Some variability in performance between synthetic targets, seedstocks, and patient samples is expected - but two orders of magnitude is a lot of variability.

3. Figure 9 does not constitute multiplex targeting. Using multiple crRNAs with a single readout (such that one cannot tell which crRNA results in cleavage) is not multiplexing. "pooling" would be a more accurate description here.

In light of these comments, I would recommend that the authors reformat their paper to streamline the results shown and resubmit to a different journal.

We thank the reviewers for their helpful suggestions, which have improved our manuscript.

Reviewer #1 (Remarks to the Author):

In this study, Ooi et al. undertook to produce a version on the CRISPR Cas12a-based SARS-CoV-2 detection assay that would be impervious to the variations in the viral RNA sequence that are bound to occur over time. In the process, they also tested and adjusted a number of parameters of the LAMP amplification step and the detection step itself. By combining a more mismatch-tolerant version of AsCas12a with two guide RNAs targeting a pair of sequences in the viral S gene, they largely accomplished their goal, while matching existing assays in sensitivity and specificity. In doing so, they maintained a clear focus on the needs of clinics and testing centers regarding simplicity, speed and cost.

Although my expertise does not lie in the area of clinical testing, this seems to me like a valuable contribution. The extensive testing of multiple variables was all done in a single lab, as far as I can see, so the robustness of the conclusions will still have to be tested more widely. Even if the parameters defined here do not precisely replicate elsewhere, important variables and approaches have been explored.

We thank the reviewer for the kind comments. The validations on clinical RNA samples were performed in the clinic by hospital staff in our absence.

The presentation is quite clear, but I would suggest a few modifications.

- Move Supplementary Figure 15 to the main text and include the locations of the swarm primers. This will help readers follow the text in several places, particularly where the various amplification primers are being altered.

As requested, we have moved the schematic to the main text (Fig. 3a) and noted in the figure legend that the swarm primers have the same sequences as F1c and B1c.

- It seems logical to put the sections on multiplexing LbCas12a and enAsCas12a together, rather than separating them with the sections on LAMP variables. The latter sections should probably precede the LbCas12a section.

To avoid confusion from readers, we have streamlined our manuscript to focus mainly on enAsCas12a.

- Provide statistics (Avg, StdDev) on the dipstick assays, particularly in cases where the signal above background is small – e.g., Figures 6b, 9f, 11e and S19c. This would increase confidence in those lower signals that are not obvious by eye.

Similar to the reviewer, we were concerned with cases where the test bands might appear faint and be subjected to conflicting interpretations. This motivated us to improve our assay such that positive cases become more clear-cut and distinct from negative cases. We largely succeeded by using hybrid DNA-RNA guides in our assay instead of normal gRNAs (Fig. 7) and by making sure that we utilized all the LAMP products in the downstream Cas detection step (Fig. 8). Now we find that the results from our dipstick experiments are more obvious, where samples containing the virus always give clear, unambiguous test bands. Furthermore, we developed a mobile phone app that takes a dipstick image as input and automatically quantifies the band intensities (Supplementary Fig. 49).

- In the results section, the authors mention the advantage of relying on the commercially available LbCas12a, and they go so far as to demonstrate its use on mismatched viral templates. Then after an extensive section on optimizing enAsCas12a, they never return in the Discussion to how one might use the Lb enzyme when the enAs enzyme is not available. A concluding comparison of the two would be useful.

As mentioned above, we have focused our manuscript on enAsCas12a, so that the reader will not be confused.

Reviewer #2 (Remarks to the Author):

In this manuscript, Ooi et al. tested the specificity of 5 different CRISPR/Cas12a variants (2 orthologs and 3 engineered AsCas12a) using the activators with mismatches. They then selected and combined two different orthologs of CRISPR/Cas12a to intentionally create a less-specific test for COVID-19 that tolerates mismatches and can detect the mutated spike protein genes of SARS-CoV-2 RNA. The authors combined an isothermal RT-LAMP step with CRISPR and reported a sensitivity of 20 copies/uL in a fluorescence-based assay for detecting the S gene. A lateral flow assay was also tested and validated with a handful of patient samples.

The sensitivity of our test is 20 copies per reaction or 0.8 copies per microliter (using synthetic SARS-CoV-2 RNA).

Novelty:

While similar fluorescence-based and paper-based tests using RT-LAMP+CRISPR/Cas12a tests have been developed and specificity of various CRISPR/Cas12a systems have been individually reported, the authors performed parallel comparisons of these variants for SARS-CoV-2 genes. They combined multiple low specificity variants of Cas12a to be able to detect both the wild-type and the mutated version of the test with similar efficiency. The authors performed optimization and analysis with different regions of SARS-CoV-2 genome to develop a COVID-19 test.

We thank the reviewer for the nice summary of our work.

Major concerns:

1. I commend the authors on having doctors perform the blind test (see reporting summary).

We thank the reviewer for the encouraging comment.

The authors tested their assay in a 6 positive and 6 negative patient samples and therefore it is not recommended to conclude 'clinically-validated' (see abstract). For instance, the FDA recommends testing at least 30 positive/30 negative samples as a guideline. Testing additional clinical samples will improve the robustness of the assay. I highly recommend testing additional samples.

We apologize for the use of the phrase “clinically validated”. Instead, we now write that we have evaluated our assay with clinical samples. We have also tested additional samples with our test. In total, we applied our assay on clinical samples from 72 patients with COVID-19 infection and 57 healthy individuals (Fig. 6d, Fig. 8i, and Fig. 9d).

2. “Nevertheless, while further work is needed to improve its clinical sensitivity, our assay is already an improvement over the DETECTR and Sherlock platforms in terms of analytical sensitivity and robustness against variant nucleotides.” In the table 1 (or in a separate table), the authors should include the known test performance data of currently available methods such as qPCR, SHERLOCK, and DETECTR? The reported LOD is 20 copies per reaction. Please include LOD in copies/uL of patient sample for direct comparison. How does it truly compares with the DETECTR (Broughton et al., Nature Biotech, 2020). If the sensitivity is lower than reported, authors cannot claim an improved sensitivity. Furthermore, as authors commented in

Lines 566-569 “Hence, our results indicate that despite its high analytical sensitivity of 20 copies per reaction, our VaNGuard assay appears to exhibit a lower clinical sensitivity possibly because the matrix of clinical samples may exert some inhibitory effect on the assay enzymes.”

Missing information about the patient samples. What sample type (nasal swabs, saliva) was chosen for this study? Since the true sensitivity might be even lower in the matrix and the true LOD should be calculated in the same matrix by pooling negative samples.

We thank the reviewer for these helpful remarks. We have now expanded the table (now known as Supplementary File 1) to include the test performance data of all currently available CRISPR-based methods (as of 18 December 2020), including analytical sensitivity, clinical sensitivity using RNA samples, and clinical sensitivity using crude (unpurified) samples. In addition, we have provided the sensitivity of our assay in both copies per reaction and copies per microliter. Overall, our VaNGuard test exhibits comparable, if not better, sensitivity than most other methods. For example, with synthetic RNA, the sensitivity of DETECTR is 10cp/μl, while the sensitivity of VaNGuard is 0.8cp/μl.

For our manuscript, we have organized the main text such that Figures 1-8 are focused on purified RNA samples, while Figures 9-10 are focused on crude (unpurified) samples. We hope that this organization will be clearer to readers.

3. Since the virus mutations and viral RNA editing in the host are indeterministic in nature and therefore it is not possible to claim that the CRISPR/Cas12a combinations would still be functional with the same efficiency after the virus mutates. Please clarify.

Yes, we agree with the reviewer’s comment. Unless we test all possible mutations and RNA editing events (which is not feasible), we can never know for sure. We have added this caveat in the discussion section of our manuscript: “further work is needed to determine how applicable the reported strategies are to other combinations of viral mutations and RNA editing events.”

4. In general, decreasing the specificity increases the sensitivity. However, combining different variants of Cas12a with different sensitivities can further lower the sensitivity of detection as shown in Figure 5. How do authors plan to maintain good sensitivity?

We are not combining different variants of Cas12a. The (old) Fig. 5 shows the different variants tested separately. Moreover, in our final assay, we are using only enAsCas12a.

5. The authors included GISAID dataset for inclusivity analysis (Supplementary Dataset 1). However, to confirm the true applicability of VaNGuard for detecting mutated samples, the inclusivity testing should be performed with at least a couple of different isolates.

We evaluated our assay on two different real-life mutations that had been found in multiple isolates from around the world, namely S254F (missense mutation) and N234N (silent mutation) (Fig. 4f, 4g, 5b, 5d, 6b, and 8g).

6. RT-LAMP is prone to false-positives and cross-contamination as shown by the authors (SI Fig. 27). Please indicate how the authors tried to resolve this issue. It is unclear from the figures.

We apologize for the lack of clarity. Our message in the discussion section is that we need to be careful of assays that rely only on RT-LAMP. In our work, the CRISPR detection module adds a specificity check to resolve the issue of potential false positives from RT-LAMP. We have added the following sentence in our manuscript to clarify: "Hence, the Cas detection step provides a valuable specificity check that rules out these undesirable false positives."

Minor concerns:

7. There are some key CRISPR-based COVID-19 tests missing from table 1 and I recommend adding them for completeness. These include: SHERLOCK (Cas13a-based, the first one to get an FDA EUA), MB-Qdot CRISPR (Cas12a-based), CRISPR-ENHANCE (Cas12a-based), SHINE (similar to CARMEN, Cas13a-based), CRISPR-CADD (dCas9-based), IMPACT chip (Cas12a-based).

We thank the reviewer for pointing out the CRISPR tests that we have missed. In our Supplementary File 1 (old Table 1), we have included SHERLOCK, CRISPR-ENHANCE, and SHINE.

We did not include MB-Qdot CRISPR, CRISPR-CADD, and the IMPACT chip because there was no work on SARS-CoV-2 in these papers. The application presented in the MB-Qdot CRISPR and the IMPACT chip papers was the African swine fever virus, while the primary application presented in the CRISPR-CADD paper was human papillomaviruses.

8. Is the enAsCas12a same as the E174R/S542R/K548R version. Please indicate that in the manuscript?

Yes, enAsCas12a is the same one as the E174R/S542R/K548R version and we have mentioned it in our revised manuscript.

9. In the reporting summary, the statistics section with statistical test information is missing. In fact, most figures are missing the statistical analysis.

We have strengthened our work by adding the necessary statistical analysis in our revised manuscript.

10. Line189 "There was no cross-reactivity for SARS-CoV or MERS-CoV."
Lines 204-205 "Importantly, no cross-reactivity for SARS-CoV or MERS-CoV was observed regardless of the Cas12a nuclease used."
Lines 225-226 "There was again no cross-reactivity for SARS-CoV and MERS-CoV. "
There are too much verbatim and no citing of the figures. The authors actually performed the bioinformatic analyses (Fig. S5) and experimental confirmation (Fig. 9a) for SARS-CoV and MERS-CoV. Please remove duplicate sentences and add proper call-outs to the figures.

We apologize for the lack of clarity. As requested, we have removed duplicate sentences and added proper call-outs to the figures.

11. Also, include the RT-LAMP primers in the alignment analysis and add it to the SI section. I also recommend aligning the sequences with all the other human coronaviruses HCoV-229E, HCoV-OC43, HCoV-NL63, and HCoV-HKU1 bioinformatically.

As requested, we have provided the sequence alignment analysis of our LAMP primers in the SI document (Supplementary Fig. 18). We have also included the sequences of HCoV-229E, HCoV-OC43, HCoV-NL63, and HCoV-HKU1 in the analysis.

12. Line: 260-262 "Strikingly, we now observed that LbCas12a exhibited the best overall tolerance for mismatches, while AsCas12a and its engineered variants became more sensitive to the mismatches."

Why was AsCas12a chosen if the LbCas12a was the best? Please add to the discussion.

We apologize for the lack of clarity. In that old section (containing lines 260-262), we paired the S1 gRNA with S2 gRNA to help LbCas12a handle variant nucleotides at the target site. But the S1 gRNA does not work with AsCas12a or its engineered variants. Hence, for the specific case of S1+S2 gRNAs only, LbCas12a performed the best. In other situations however, LbCas12a performed poorer than enAsCas12a.

To avoid confusion from readers, we have streamlined our revised manuscript to focus primarily on the overall best performing enzyme, enAsCas12a.

13. Line 159-160: "Further increase in temperature from 65oC to 68oC caused a deterioration in the performance of our CRISPR-Dx (Supplementary Fig. 4)".

Did the authors mean RT-LAMP instead of CRISPR-Dx?

We apologize for the lack of clarity. In our revised manuscript, we have been more careful to distinguish between RT-LAMP and the downstream CRISPR detection module.

14. Figure 2: Please clearly indicate on the plot that the 62oC, 65oC represents RT-LAMP temperatures.

The old Figure 2 is no longer in our revised manuscript.

15. For Cas12a, I would call 'crRNA' instead of 'gRNA'. Interestingly, the term 'crRNA' is not even mentioned in the entire manuscript.

We debated about whether to use crRNA or gRNA. Since Cas12a does not need a tracrRNA, its crRNA is effectively its guide RNA or gRNA. (In contrast, the gRNA of Cas9 consists of both the crRNA and the tracrRNA.) In the end, we decided to follow the nomenclature in the DETECTR publication (PMID: 32300245), where gRNA was used and crRNA was also not mentioned. Moreover, in the original Cpf1 paper from Feng Zhang's lab (PMID: 26422227), the authors referred to the crRNA of Cpf1 as its "guide RNA" (please see below).

Reviewer #3 (Remarks to the Author):

This paper by Ooi, Liu, Tay, Teo, and Kaewsapsak et al. describes a CRISPR-based diagnostic assay for COVID-19 called VaNGuard. The assay uses an engineered Cas12, which is more tolerant to mismatches than the wildtype Cas12. The authors demonstrate that their assay is sensitive and specific for SARS-CoV-2, and test a small number of patient samples.

We thank the reviewer for the succinct summary.

Although some mutations have occurred in the SARS-CoV-2 genome, there are only a handful of sites where the minor allele frequency is high (as of September 18, approximately 10 sites have a minor allele frequency > 10%). Thus, the problem that the authors are setting out to address is relatively minor, consisting of improved performance on hypothetical viral sequences that may at some point emerge in the future.

While we agree that potential viral mutations are hypothetical, we would argue that the problem is a real one. The recent SARS-CoV-2 variant that is spreading rapidly throughout the United Kingdom, VUI – 202012/01, underscores the reality that mutations can and will appear and that we need to be prepared for them.

In addition, although our current manuscript is on COVID-19, we believe that the strategies presented here, such as the use of enAsCas12a and hybrid DNA-RNA guides, can be applied to detect other infectious agents and may be useful in future pandemics.

This assay, as described, is very similar to the previously published DETECTR assay for SARS-CoV-2. The authors have made some minor changes to the assay (modifying individual nucleotides in the primer sequences, changing the RT-LAMP temperature, and using different Cas12 ortholog & guide RNAs).

In our revised manuscript, we have presented more substantial data to demonstrate why our assay is different and superior to the published DETECTR assay. Supplementary Table 1 summarizes the test performance data of all currently available CRISPR-based methods (as of 18 December 2020). Specifically, compared to DETECTR, our assay not only exhibits better analytical sensitivity but is also faster, requires fewer assets to perform (1 heat block vs 2), and has a built-in human internal control within the same reaction tube. We have also applied our assay on crude clinical samples directly without any RNA purification.

Major comments:

1. The DETECTR assay published by Broughton et al. already uses multiple crRNAs targeting SARS-CoV-2 (N and E genes), so the concept of using multiple crRNAs is not novel. There is a broader lack of novelty in this paper, as most of the results shown in the figures are incremental or minor modifications of existing protocols.

In the DETECTR assay, “reactions were performed independently for N gene, E gene and RNase P”. In contrast, we have two gRNAs within the same reaction tube.

Overall, we have performed further work to enhance our assay substantially. First, we have shown that the use of hybrid DNA-RNA guides enhances the kinetics of the CRISPR detection module. Second, with the implementation of a simple Proteinase K and heat

treatment step, our assay can now accept crude clinical samples directly without any RNA extraction. Third, we have shown that enAsCas12a exhibits a surprising robustness to reaction temperature. This property enables us to perform RT-LAMP and the Cas detection step at the same temperature (and thus our assay only requires a single heat block). Fourth, we incorporated a human internal control into the same reaction tube as the COVID-19 test. Collectively, we hope that these improvements, together with our earlier reported strategies to boost the robustness of our assay against single nucleotide variations, are sufficiently novel.

2. The assay has reduced performance on viral seedstocks (Fig. 11d) relative to synthetic targets, by at least 1 order of magnitude (10-fold). There is a further reduction of sensitivity in Fig. 11e, where the authors are only able to detect patient samples down to ~190 copies. The authors speculate that this could be due to clinical matrix, but the samples consist of extracted RNA so this is not possible. Some variability in performance between synthetic targets, seedstocks, and patient samples is expected - but two orders of magnitude is a lot of variability.

We have performed more careful evaluations of our VaNGuard test.

- The analytical sensitivity of our assay for pure synthetic RNA is 20 copies per reaction.
- Its clinical sensitivity for purified RNA patient samples is 50 copies per reaction.
- Its sensitivity for SARS-CoV-2 virions spiked into clinically negative UTM is 100 copies per reaction.
- Its clinical sensitivity for crude clinical samples (NP swabs) is 1000 copies per reaction. As noted in our manuscript, the swabs were collected in 3ml UTM and we only used 2 μ l of it for testing. Hence, we believe that this sensitivity will be better if less collection medium is used.

Overall, as the sample complexity increases, we do observe an expected decline in sensitivity. Nevertheless, the fold-change in sensitivity between pure synthetic RNA and SARS-CoV-2 virions is less than one order of magnitude.

3. Figure 9 does not constitute multiplex targeting. Using multiple crRNAs with a single readout (such that one cannot tell which crRNA results in cleavage) is not multiplexing. "pooling" would be a more accurate description here.

We thank the reviewer for the comment. "Pooling" might also not be a good way to describe the approach, as clinicians refer to it as pooling several samples together in a single test to save reagents. Hence, we have used the term "two-gRNA" instead.

In light of these comments, I would recommend that the authors reformat their paper to streamline the results shown and resubmit to a different journal.

As suggested, we have streamlined our manuscript by focusing on the novel results and demonstrating how our assay possesses unique advantages over existing CRISPR tests. We hope the reviewer will now find our revised work to be of sufficient quality for the journal.

Reviewers' Comments:

Reviewer #2:

Remarks to the Author:

The authors addressed most of the previous concerns. They provided supporting data, including additional clinical samples.

1. The authors listed specificity as 100%, but the lateral flow data in patient samples suggest several false negatives and false positives in figures 8i-j and 9d-f based on the overlapping ratios. For example, several positive and negative patient samples have ratios between 0.07-0.10. It is unclear how the authors chose a threshold value for validation. The authors should correctly calculate and report both false negatives and false positives. Furthermore, determine and report positive predictive values and negative predictive values.

Reviewer #3:

Remarks to the Author:

I thank the authors for their revisions to this work. As more and more SARS-CoV-2 variants are now spreading through the population, the need to pool crRNAs is becoming more evident.

I also appreciate how the authors have improved the paper, particularly the clinical testing component, which was one of my biggest concerns with the initial version of the paper.

Based on these improvements, I believe the work is now suitable for publication. However, I would urge the authors to consider if there are ways to further streamline the presentation of their work. There are ten main-text figures at the moment.

Reviewer #2 (Remarks to the Author):

The authors addressed most of the previous concerns. They provided supporting data, including additional clinical samples.

We thank the reviewer for the positive comments.

1. The authors listed specificity as 100%, but the lateral flow data in patient samples suggest several false negatives and false positives in figures 8i-j and 9d-f based on the overlapping ratios. For example, several positive and negative patient samples have ratios between 0.07-0.10. It is unclear how the authors chose a threshold value for validation. The authors should correctly calculate and report both false negatives and false positives. Furthermore, determine and report positive predictive values and negative predictive values.

We apologise for the lack of clarity. A ratio of less than 0.15 was considered to be negative in our test. This threshold was chosen based on all the lateral flow assays we had performed over the course of the project. Hence, for the purified RNA samples (Fig. 8i-j), our VaNGuard test gave 0 false positives and 8 false negatives (RP6, RP45-51). In addition, for the direct patient samples (Fig. 9d-f), our test gave 0 false positives and 4 false negatives (DP16, DP18, DP20, and DP21). We have reported the number of false positives and false negatives in the figure legends.

Specificity is defined as:

$\text{number of true negatives} / (\text{number of true negatives} + \text{number of false positives})$.

Since the number of false positives is 0, our test specificity is 100% as reported.

We thank the reviewer for suggesting the use of positive predictive value (PPV) and negative predictive value (NPV). Our test PPV is 1 (or 100%), which we have included in the abstract. Nevertheless, we find the NPV to be misleading because it depends on the relative number of “easy” samples (high viral loads) and “difficult” samples (low viral loads) that have been used for testing. If one chooses to test only samples with high viral loads (i.e. low Ct values), it will be easier to obtain a NPV close to 1. Hence, we prefer to report the Ct threshold or the copy number threshold where our test will work or fail.

Reviewer #3 (Remarks to the Author):

I thank the authors for their revisions to this work. As more and more SARS-CoV-2 variants are now spreading through the population, the need to pool crRNAs is becoming more evident.

I also appreciate how the authors have improved the paper, particularly the clinical testing component, which was one of my biggest concerns with the initial version of the paper.

Based on these improvements, I believe the work is now suitable for publication. However, I would urge the authors to consider if there are ways to further streamline the presentation of their work. There are ten main-text figures at the moment.

We thank the reviewer for the positive comments.

We agree that our manuscript is longer than normal. Nevertheless, each main-text figure provides a different key message to the reader. Hence, we think that it will be clearer to maintain all the figures as-is.